# Molecular circadian rhythms are robust in marine annelids lacking rhythmic behavior

N. Sören Häfker[1,2]*, Laurenz Holcik[1,3,4], Audrey M. Mat[1], Aida Ćorić[1,3], Karim Vadiwala[1¤a], Isabel Beets[5], Alexander W. Stockinger[1,3], Carolina E. Atria[6,7], Stefan Hammer[1¤b], Roger Revilla-i-Domingo[1,6,7], Liliane Schoofs[5], Florian Raible[1], Kristin Tessmar-Raible[1,2,8]*

**1** Max Perutz Labs, University of Vienna, Vienna BioCenter, Vienna, Austria, **2** Alfred Wegener Institute Helmholtz Centre for Polar and Marine Research, Bremerhaven, Germany, **3** Vienna BioCenter PhD Program, Doctoral School of the University of Vienna and Medical University of Vienna, Vienna, Austria, **4** Center for Integrative Bioinformatics Vienna, Max Perutz Labs, University of Vienna, Medical University of Vienna, Vienna, Austria, **5** Division of animal Physiology and Neurobiology, KU Leuven, Leuven, Belgium, **6** Department of Neuro- and Developmental Biology, University of Vienna, Vienna, Austria, **7** Research Platform Single-Cell Regulation of Stem Cells, University of Vienna, Vienna, Austria, **8** Institute for Chemistry and Biology of the Marine Environment (ICBM), School of Mathematics and Science, Carl von Ossietzky Universität Oldenburg, Oldenburg, Germany

¤a Current address: Biosynth GmbH, Campus Vienna Biocenter 2, Vienna, Austria
¤b Current address: Institut für Diskrete Mathematik, TU Graz, Graz, Austria
* soeren.haefker@awi.de (NSH); kristin.tessmar@mfpl.ac.at (KT-R)

**Data Availability Statement:** Relevant data are within the paper and its Supporting Information files. The large primary data files are deposited at

## Abstract

The circadian clock controls behavior and metabolism in various organisms. However, the exact timing and strength of rhythmic phenotypes can vary significantly between individuals of the same species. This is highly relevant for rhythmically complex marine environments where organismal rhythmic diversity likely permits the occupation of different microenvironments. When investigating circadian locomotor behavior of *Platynereis dumerilii*, a model system for marine molecular chronobiology, we found strain-specific, high variability between individual worms. The individual patterns were maintained for several weeks. A diel head transcriptome comparison of behaviorally rhythmic versus arrhythmic wild-type worms showed that 24-h cycling of core circadian clock transcripts is identical between both behavioral phenotypes. While behaviorally arrhythmic worms showed a similar total number of cycling transcripts compared to their behaviorally rhythmic counterparts, the annotation categories of their transcripts, however, differed substantially. Consistent with their locomotor phenotype, behaviorally rhythmic worms exhibit an enrichment of cycling transcripts related to neuronal/behavioral processes. In contrast, behaviorally arrhythmic worms showed significantly increased diel cycling for metabolism- and physiology-related transcripts. The prominent role of the neuropeptide pigment-dispersing factor (PDF) in *Drosophila* circadian behavior prompted us to test for a possible functional involvement of *Platynereis pdf*. Differing from its role in *Drosophila*, loss of *pdf* impacts overall activity levels but shows only indirect effects on rhythmicity. Our results show that individuals arrhythmic in a given process can show increased rhythmicity in others. Across the *Platynereis* population, rhythmic phenotypes exist as a continuum, with no distinct "boundaries" between rhythmicity and arrhythmicity. We suggest that such diel rhythm breadth is an important

the Dryad online repository (doi:10.5061/dryad. 31zcrjdnq).

**Funding:** This work was supported with a Helmholtz distinguished professorship by the Alfred Wegener Institute Helmholtz Centre for Polar and Marine Research;the FP7 Ideas: European Research Council, ERC Grant Agreement 792 #337011, Wissenschaftsfond FWF, SFB F78 and H2020 European Research Council, ERC Grant Agreement #819952 to KT-R; by the Wissenschaftsfonds FWF, Lise-Meitner fellowship M2820 to NSH; by the KU Leuven Research Council, C16/19/003 to IB and LS; the Research Foundation Flanders, FWO G0B5322N to IB; the Wissenschaftfond FWF, #I2972 and SFB F78, FP7 Ideas: European Research Council, #260564 to FR; and the H2020 European Research Council, Marie 796 Skłodowska-Curie Grant Agreement #847548 to AMM. The funders had no role in study design, data collection and analysis, decision to publish, or preparation of the manuscript.

**Competing interests:** The authors have declared that no competing interests exist.

**Abbreviations:** CHO, Chinese Hamster Ovary; DD, constant darkness; EST, Expressed Sequence Tag; GO, Gene Ontology; IHC, mmunohistochemistry; KS-test, Kolmogorov–Smirnov test; LD, light/dark cycle; MALDI-TOF-MS, matrix-assisted laser desorption/ionization time-of-flight mass spectrometry; MMP, matrix metalloproteinase; ORF, open reading frame; PDF, pigment-dispersing factor; RVD, repeat-variable di-residue; TALEN, transcription activator-like effector nuclease; ZT0, Zeitgeber time0.

biodiversity resource enabling the species to quickly adapt to heterogeneous or changing marine environments. In times of massive sequencing, our work also emphasizes the importance of time series and functional tests.

## Introduction

Life on earth has been evolving in the presence of numerous environmental cycles. This is particularly prominent in marine environments [1,2], where rhythms even exist in the deep sea [3]. In the oceans, biological rhythms can be the major factors shaping ecosystem structure and productivity, and they significantly contribute to biogeochemical cycling [4–6]. However, compared to the terrestrial environment, our understanding of the factors and mechanisms driving marine rhythmicity is still severely limited [2]. This is particularly concerning in the context of climate change as it is largely unknown how marine timing systems and associated species fitness on the individual and population level will be affected by changing environmental conditions [2,7].

Rhythmic adaptations on the level of behavior, physiology, and gene expression are central in shaping an animal's interaction with the abiotic and biotic environment. Many of these rhythms are under the control of endogenous timing mechanisms (clocks) [7,8]. Among those the best studied are circadian clocks. These inner oscillators enables organisms to synchronize (entrain) to, and to anticipate the 24-h day/night cycle [9], thereby significantly contributing to fitness [10,11]. Like all biological processes, approximately 24-h rhythmicity varies between individuals, as prominently illustrated by human chronotypes [12].

Studies in terrestrial organisms have most frequently identified changes to the core circadian clock as cause of the variance: Human chronotypes are linked to differences in the endogenous core circadian clock period length [13]. Differences in clock gene expression rhythms correlate with caste-specific activity phenotypes in ants [14]. The marine midge *Clunio* shows habitat- and strain-specific temporal niche adaptations of circadian emergence timing that are linked to splice variants in a *calcium-calmodulin-dependent kinase*. These splice variants impact on core clock protein phosphorylation, thereby likely affecting circadian period length [15]. Several studies in drosophilid flies identified cellular and molecular mechanisms underlying latitude-dependent adaptations of species- and population-specific patterns of diel activity. Also in these cases, expression of circadian clock genes and changes in the core circadian clock network via the neuropeptide pigment-dispersing factor (PDF) correlated strongly with the behavioral rhythmicity phenotype [16–21]. Specifically, comparisons of fly strains and species from different latitudes indicate that behavioral rhythmicity depends on where in the brain different circadian core clock genes are expressed, on circadian neuronal network interconnectivity, and on differences in clock gene alleles [16–24]. These results fit the common assumption that circadian rhythm power (i.e., rhythm amplitude and precision) of a particular read-out like locomotor activity is representative for the rhythmicity of the entire organism and directly interconnected to the function of the core circadian oscillator. However, integrative studies that explicitly test this assumption are rare. Of note, some drosophilid fly species that inhabit high latitudes become behaviorally arrhythmic while maintaining functional molecular circadian clock oscillations. Based on these results it has been suggested that the resulting behavioral arrhythmicity occurs by uncoupling the circadian clock from its output pathways [20]. Similar conclusions were drawn from a study on the Svalbard ptarmigan, an Artic resident bird [25]. Furthermore, multiple studies of diurnal versus nocturnal organisms

suggest that this temporal niche switch of behavior occurs downstream of the core circadian oscillator [26,27]. Together, these studies strongly indicate that circadian behavioral rhythms might not always be representative of core circadian oscillator function and, hence, general circadian organismal rhythmicity.

Behavioral variability can arise seemingly at random but can also be consistent over time, then referred to as "personality," "behavioral syndrome," or "coping style" [28,29]. While often such behavioral "personalities" are intuitively associated with vertebrates, they have in fact been documented in a variety of organisms from invertebrates to mammals [28–30]. Due to the relatively short life time of *Drosophila melanogaster*, individual variance of approximately 24-h rhythms can only be explored over a limited time. However, many invertebrates live much longer, and especially in the context of understanding the adaptation potential of circadian clocks in populations, comprehending the extent and source of their individual variability is critical. The understanding of how differences in circadian behavioral rhythms across individuals and populations interlink with their molecular features is still highly limited. This applies especially to marine organisms, despite the fundamental importance of biological rhythms and clocks for marine ecosystems [2].

The marine polychaete *Platynereis dumerilii* is a molecularly slowly evolving annelid that has emerged as a functional model for marine chronobiology with a life cycle representative for various marine invertebrates [31–34]. After a planktonic larval phase during which animals get dispersed by currents, the worm switches to a benthic lifestyle that ends with a metamorphic maturation into a free-swimming spawning form [31]. It inhabits temperate to tropical shores worldwide, typically at depths of 0 to 10 m [35,36]. The experimental culture used here derives from the Mediterranean Sea, where the worm is part of the ecologically important *Posedonia* sea grass meadows [34].

Here, we explore approximately 24-h behavioral diversity of *Platynereis* strains and individuals, its correlated temporal transcriptomes, and the role of PDF, an important neuropeptide for circadian behavioral rhythmicity in *Drosophila* [16,17,19]. We tested whether the behavioral circadian variability in *Platynereis* occurs at random or represents a reproducible individual trait, and how this might be reflected on the molecular level. Worms displayed strain-specific characteristics as well as strong and reproducible interindividual variability in diel behavior, even among coraised siblings. A diel transcriptome comparison of heads from behaviorally rhythmic versus arrhythmic wild-type worms showed that these phenotypic differences correlate with the types of diel cyclic transcripts, but not their overall number. Differences were especially evident for transcripts involved in neuronal signaling versus metabolic processes. In contrast, cycling of core circadian clock genes was identical between the different behavioral groups, indicating that in *Platynereis* natural variants, the level of behavioral versus metabolic rhythm power is controlled by genes downstream of the core circadian clock (e.g., genes of neuronal functions versus metabolic enzyme genes).

Given that in *Drosophila* PDF has central roles in circadian behavioral rhythms [16,17,19], we tested the effects of *pdf* knockouts in *Platynereis*. The phenotype was remarkably different from *Drosophila*, as there was no direct effect on circadian period or rhythm power of locomotor activity after several outcrosses. Instead, *pdf* mutants showed overall more activity, which occasionally had indirect effects on rhythm power under light/dark cycles. Thus, our work also provides new perspectives on the evolution of PDF. While PDF's role in general behavioral control is conserved across major invertebrate groups, its role in circadian control outside insects might not be conserved.

Considering the complexity of the cyclic environment in marine habitats [1,2], unraveling the molecular processes shaping phenotypic rhythms and their diversity will be important to

understand how marine species adapt to these conditions, and how individuals and populations cope with the ongoing shifts in habitat conditions due to climate change.

## Results

### *Platynereis* circadian behavioral rhythmicity differs between and within strains but is largely reproducible on the individual level

In our previous studies, we repeatedly observed a high diversity in circadian behaviors of immature and premature *P. dumerilii* bristle worms [32,37,38]. While some of these differences can be explained by interactions of the worms' plastic circadian-circalunidian clock, circalunar clock, and light [37,38], we wondered about the remaining variance at identical lunar phases and light conditions. This prompted us to systematically investigate the worms' diversity in circadian locomotor rhythmicity patterns at the new moon phase. Please note that in the results below, we use the term "rhythmicity/arrhythmicity" to specifically refer to behavior, while we use the term "cycling" in the context of gene expression.

We chose 3 worm strains (VIO and PIN: collected from the Bay of Naples, Italy more than 50 years ago and continuously incrossed over the past 14 years; NAP: newly collected from the Bay of Naples in 2009 and maintained as a closed culture without targeted incrossing) [34,39]. We recorded locomotor behavior of immature/premature sibling worms for 4 d of 16 h:8 h light/dark cycle (LD) followed by 4 d of constant darkness (DD) (Figs 1A and S1). All strains were generally nocturnal and showed little overall interstrain differences in morning/afternoon/night activity (Fig 1B), as reported previously [37]. When we analyzed rhythmicity (20-h to 28-h period range) under LD and DD in more detail, we noticed that strain-specific rhythmicity patterns and rhythm power varied significantly between strains, particularly under LD conditions (Fig 1A–1D). Additionally, all strains displayed pronounced interindividual variability in rhythm power that is best described as a behavioral continuum ranging from highly rhythmic to fully arrhythmic worms (Figs 1C, 1D and S2 and S1A Table). Importantly, arrhythmic individuals were not immobile, but rather showed a lower, but continuous level of activity throughout time, which becomes visible with a different y-axis scale (compare S2 and S3 Figs, e.g., for PIN #3,4,7,20,22,25). For the VIO strain, the lower rhythm power in LD (Fig 1C) is possibly related to the Lomb–Scargle rhythm analysis (see Materials and methods), which assigns lower rhythm power values to the short activity bursts at dusk and/or dawn (S2C Fig).

To investigate more closely the observed individual diversity, we performed repetition runs on a total of 30 worms (PIN *n* = 12, NAP *n* = 5, VIO *n* = 13) 1 to 2 month after the initial recordings. Repetitions were limited, because some individuals matured before they could be reinvestigated, which leads to irreversible behavioral changes [34]. Visual inspection of the actograms suggested that the behavior of individual worms was nonrandom, i.e., arrhythmic individuals exhibited a similar behavior in the repetition run (Fig 1E, e.g., PIN #3,20,22,25), while individuals with specific activity peak times also tended to show this rhythmicity in the repetition (Fig 1E, e.g., PIN #12,15,18,19). This was particularly observable in the PIN strain and, to a lower extent, in VIO and NAP (S4 Fig). In addition to visual inspection of initial/repetition similarity, we statistically quantified the locomotor behavior. Given the complexity of the behavioral patterns, we decided to use 2 different and fully independent approaches, which differed in their level of complexity reduction.

First, we employed a binary activity overlap approach. In brief, the activity profile of each worm behavior run was transformed into a binary activity vector, separately for LD and DD light conditions. The binary activity vectors were then used to quantify the similarity between pairs of behavior runs from the same individual. For this, we calculated the "overlap activity score," defined as the number of 30-min bins at which both behavior runs showed activity

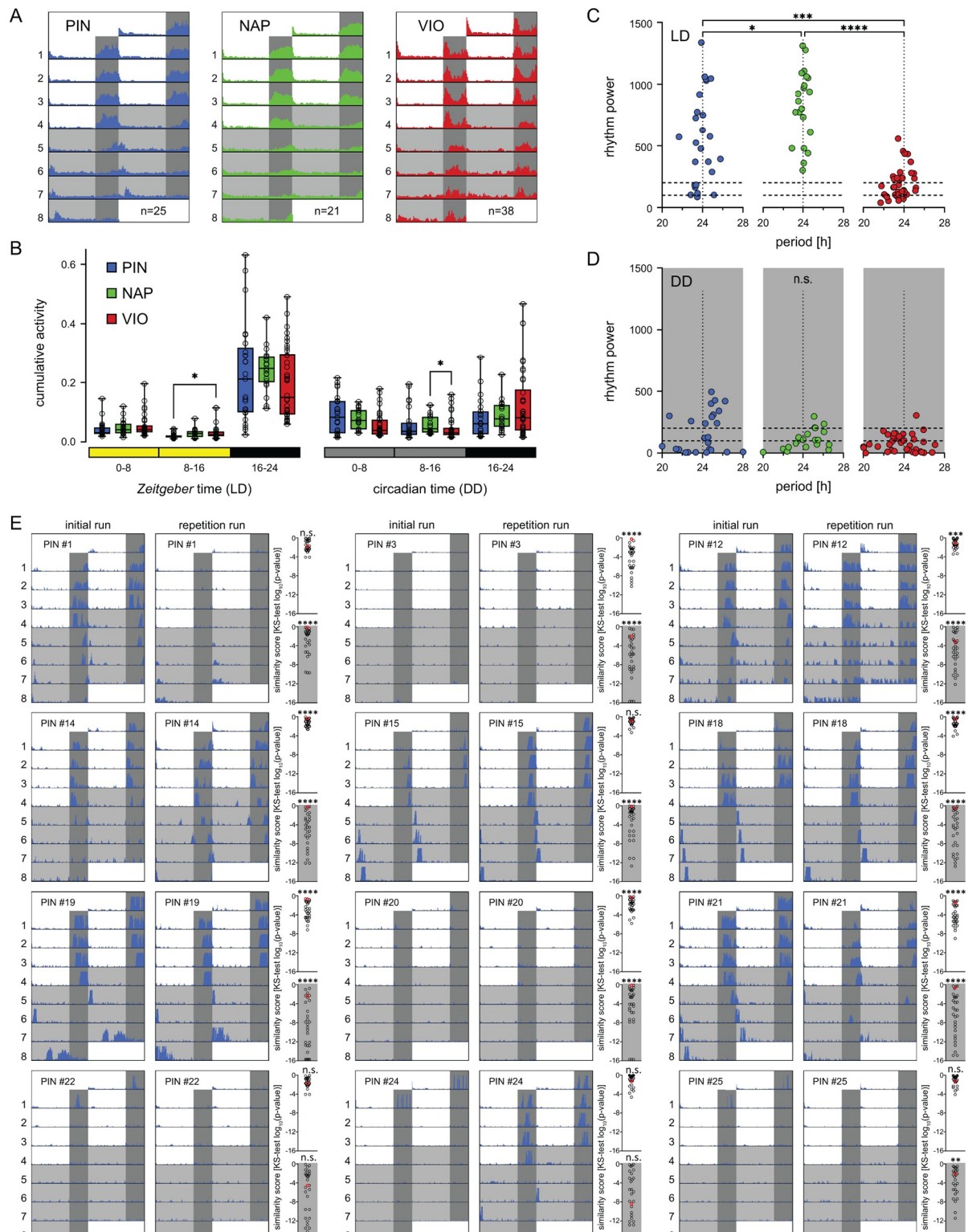

**Fig 1. Interstrain and interindividual variability in circadian behavioral rhythmicity.** The circadian locomotor activities of the 3
*Platynereis* strains PIN (blue), NAP (green), and VIO (red) are compared. (**A**) Double-plotted actograms of mean locomotor activity over 4 d
of a 16 h:8 h light/dark cycle (LD) followed by 4 d of constant darkness (DD). Individual worm actograms are provided in S2 Fig. (**B**)
Cumulative activity across early day (0–8), late day (8–16), and night (16–24) in LD and DD. Box: median with 25%/75% percentiles,
whiskers: min/max. (**C, D**) Period/power of locomotor rhythms in the 20-h–28-h range in LD and DD determined by Lomb–Scargle

periodogram. Dashed horizontal lines: thresholds for strong rhythm power (>200) and complete arrhythmicity (<100). Each circle: individual worm. Statistical differences were determined via Kruskal–Wallis ANOVA on Ranks (panels B and D) or Welch's ANOVA (panel C). Significance levels: *$p < 0.05$, **$p < 0.01$, ***$p < 0.001$, ****$p < 0.0001$. (**E**) Actograms of PIN individuals investigated in 2 consecutive runs (initial/repetition). Scatter plots for LD and DD (gray background) show $\log_{10}(p$-values) of Kolmogorov–Smirnov test (KS-test) comparisons of initial vs. repetition runs, which serve as a proxy of behavioral similarity (high $p$-value = high similarity; see Results and Materials and methods). Each initial run was compared to the matching repetition run (red circle) and to all other repetition runs (black circles, $n$ = 29). Similarity differences between same-individual repetition compared to all other repetition runs were determined via one-sample Wilcoxon signed-rank test. Significance levels (FDR-corrected): *$p < 0.05$, **$p < 0.01$, ***$p < 0.001$, ****$p < 0.0001$. #: individual worm identifier. For initial/repetition actograms of NAP and VIO strain individuals, see S4 Fig. Period/power values for panels C and D are provided in S1A Table.

(value 1 in both vectors) divided by the total number of bins where either of the 2 runs showed activity (value 1 in one or both of the vectors). For behavioral activity runs of other individuals, the mean of all activity scores was used in order to compare an equal number of data points for same-individual repetition run versus all repetition runs from other individuals. Thereby, each individual worm received a data pair of overlap activity scores for the matching and all other repetition runs in both LD ($n$ = 30) and DD ($n$ = 29). We then compared reproducibility (overlap activity scores) of runs from the same versus other individuals via one-tailed Wilcoxon matched-pairs signed-rank test. Indeed, the overlap activity scores—and, thus, reproducibility —of runs from the same individual were higher than those of the other, nonmatching worm runs (S5 Fig, LD: $p$ = 0.0007, DD: $p$ = 0.0340; see Materials and methods for details).

As a second, independent approach, we quantified behavioral reproducibility by performing Kolmogorov–Smirnov test comparisons of initial versus repetition runs, as previously used for analyses of behaviors in the marine crab *Uca* [40]. Each initial run was compared to the matching repetition run of the same worm and to the repetition runs of all other worms ($n$ = 29), which served as a control group. The resulting $p$-values were used as measure for similarity, i.e., the larger the $p$-value, the more similar the respective compared behavioral patterns. We then ranked these $p$-values and tested if the initial versus repeat runs gave significantly higher $p$-values than expected by chance when comparing to all other worm runs (Fig 1E, see plots right side of individual worm actograms, S4 Fig; for further details, see Materials and methods). Consistent with visual inspections of the actograms, PIN individuals showed significantly higher $p$-values in their initial versus repeat runs, compared to all other runs in the majority of cases, i.e., were most reproducible in their individual behaviors (Fig 1E, see plots right side of individual worm actograms). While initial versus repeated run were also in several cases similar to each other in VIO and NAP worms, this was less clear than for worms of the PIN strain (S4 Fig).

Taken together, the visual actogram analyses and 2 methods of behavioral quantifications showed significant reproducibility of individual worm locomotor behavior, especially for inbred PIN strain worms. It can certainly be stated that behavioral rhythmicity versus arrhythmicity is a nonrandom behavioral feature of individual worms. Of note, after the initial run, we kept all worms individually for identification. These culture conditions were quite different from the initial "group culture" that the worms experienced before the initial run. Some of the outliers observed in our quantitative behavioral analyses might be due to differential responses to these culture condition differences, but overall, it shows that in most cases, the circadian activity patterns in *Platynereis* are reproducible in individuals over time, irrespective of basic culture conditions.

## Core circadian clock gene transcripts cycle identically in behaviorally rhythmic and arrhythmic worms

In order to obtain insight into the molecular characteristics associated with the individually different rhythmic behavior, we investigated the diel transcriptome of behaviorally rhythmic

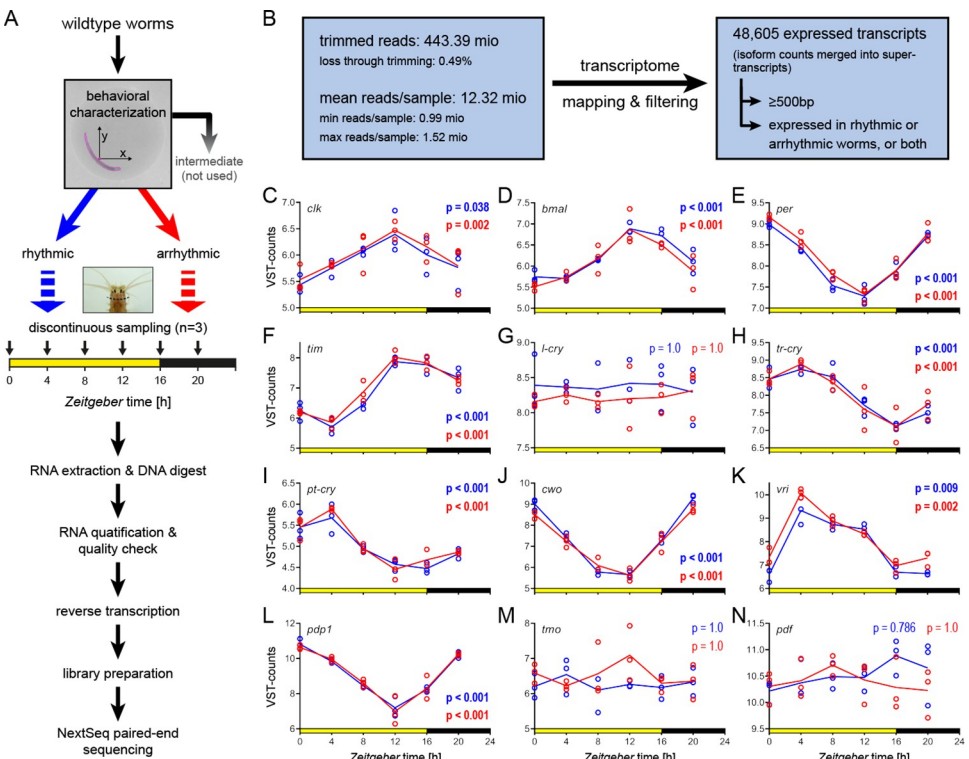

**Fig 2. DEseq analysis reveals identical core circadian clock transcript cycling in rhythmic vs. arrhythmic *Platynereis*.** (**A**) Diel locomotor behavior of PIN strain wild types was characterized as rhythmic (blue) or arrhythmic (red). Worm heads were sampled discontinuously over a 16 h:8 h LD cycle (*n* = 3 biological replicates (BR) per time point, 3 heads pooled/BR). Individual actograms are provided in S6 Fig. (**B**) Overview of the sequencing analysis pipeline. (**C-N**) Diel expression patterns of core circadian clock and neuropeptide transcripts. Expression is shown as VST-counts (see Materials and methods). Bold *p*-values indicate significant 24-h cycling as determined by RAIN analysis. Period/power values of individuals are provided in S1B Table. Raw and VST-counts are provided in S2A and S2B and S2C and S2D Table, respectively. Gene abbreviations in panels C-N and detailed RAIN results are provided in S2E and S2F Table.

and arrhythmic wild-type worms. We focused on worms of the PIN strain, as their circadian activity patterns showed the strongest within-individual reproducibility (see above and Figs 1E, S4A and S6).

Behavior of wild-type worms was recorded for 3 d of LD and 3 d of DD, and worms were characterized as either rhythmic (clear 24-h rhythmicity), arrhythmic (no rhythmicity), or intermediate (everything in between) based on Lomb and Scargle periodogram analysis and visual actogram inspection (Figs 2A, S6 and S7 and S1B Table; see Materials and methods for details). As for the initial comparison of different strains (see above), arrhythmic worms were not immobile but showed lower, evenly distributed activity throughout time (compare S6 and S7 Figs). Heads of worms identified as behaviorally rhythmic or arrhythmic were collected under LD (16 h:8 h) in 4-h intervals starting at *Zeitgeber* time0 (ZT0, lights on). To prevent possible effects of *Platynereis* circalunar rhythmicity [37], all samples (here and throughout the study) were collected at the same lunar phase around new moon (see Materials and methods). Transcriptome sequencing on a NextSeq550 System (Illumina, USA) resulted in a total of 445,553,522 raw reads (Fig 2B). Trimming/filtering caused a 0.49% read loss, and trimmed reads were mapped against an independently generated transcriptome (see Materials and methods). The mapping results were filtered to exclude transcripts that were not/almost not

expressed and/or <500 bp, resulting in 48,605 distinct expressed transcripts (S2A–S2D Table), highly consistent with previous differential expression by sequencing (DEseq) analyses [41].

We first focused on core circadian clock transcripts (Fig 2C–2N) and overall transcriptome rhythmicity. Core circadian clock transcripts of the behaviorally rhythmic worms also served as benchmark to assess if the sampling had affected transcriptome rhythmicity. Core circadian clock gene patterns (Fig 2C–2L) closely mirrored previous reports [32,37,42], emphasizing reproducibility.

To further investigate possible molecular differences in transcriptomic cycling between behaviorally rhythmic and arrhythmic worms, we identified transcripts with 24-h cyclic expression via RAIN [43]. We detected a total of 1451 transcripts (2.99% of all expressed transcripts) with 24-h cycling in rhythmic worms only: 603 transcripts, arrhythmic worms only: 554 transcripts, or both phenotypes (294 transcripts) (Fig 3A and S2E and S2F Table). Thus, the number of significantly diel cycling transcripts was comparable between rhythmic and arrhythmic worms, even though only 20% of all cycling transcripts were shared between both phenotypes (Fig 3A). For the core circadian clock and neuropeptide transcripts, RAIN analyses identified significant cycling in all of them for both phenotypes, except for *l-cry*, *tmo*, and *pdf* (Fig 2C–2N), consistent with previous results [37,42,44]. Of note, the temporal expression patterns of core circadian clock transcripts were basically identical between behaviorally rhythmic and arrhythmic worms (Fig 2C–2L). Aside from the core circadian clock transcripts, both behavioral phenotypes showed significant cycling of transcripts for *6–4 photolyase*, *r-opsin3*, *G$_o$-opsin1*, and *myocyte enhancer factor 2* (*mef2*) (S2E and S2F Table).

Beyond these approximately 20% shared cycling transcripts, the analyses suggested that behavioral arrhythmicity does not imply a generally lower number but a different set of cycling genes (Fig 3).

In principle, differences between phenotypes could arise from 2 main scenarios: (A) similar cycling patterns of the transcripts, but different amplitudes, such that the cycling is statistically significant in one phenotype, but not the other. This could arise when individual worms are still rhythmic for a given transcript over time, but the phases of this transcript would be less synchronized across different individuals. As we sampled multiple worm heads for each time point, this would—due to averaging—result in a seemingly lowered amplitude or even the loss of rhythmicity for a given transcript. (B) Different temporal patterns, with clear rhythmicity in one group, but not the other group. This would indicate that a given transcript has indeed a different temporal pattern in the respective phenotype. We thus inspected in detail how such differentially rhythmic transcripts look like. As counts for the heatmaps were normalized together across rhythmic and arrhythmic worms, expression of each transcript is directly comparable between phenotypes (Fig 3B–3D). We find evidence for both proposed types of difference (please zoom in and compare individual transcript lines in Fig 3B and 3D; for representative examples, see S8A and S8B Fig), suggesting that lower cross-individual synchronization and different temporal transcriptional/posttranscriptional processes contribute to the differences in molecular cycling between behaviorally rhythmic versus arrhythmic worms. We further expand upon the reasons for differential transcript cycling at the end of the Gene Ontology (GO) term analyses below.

## Functional categorization by Gene Ontology (GO) term analyses suggests strong metabolic rhythmicity in behaviorally arrhythmic worms

We next tested if the differences in significantly diel cycling transcript patterns correlate with differences in specific molecular/physiological processes. Thus, we categorized the cyclic transcripts for their biological functions by GO-term analysis and subsequently investigated if any

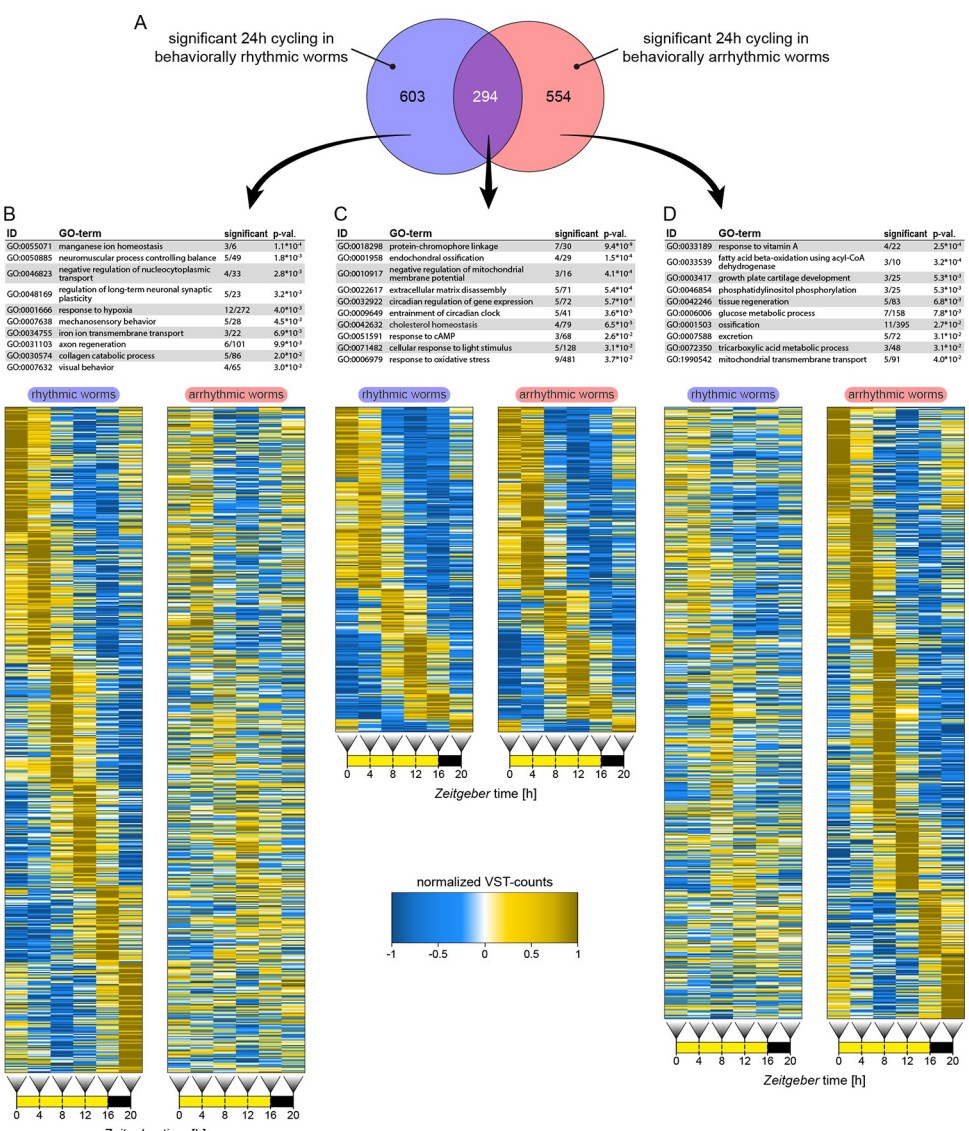

**Fig 3. Representation of cycling transcripts and GO-term annotations.** (**A**) Transcript identified by RAIN analysis with significant 24-h cycling in behaviorally rhythmic worms (blue), arrhythmic worms (red), or both. (B-D) Ten most representative biological process GO-terms for each group. Diel expression of transcripts with significant cycling in either group is plotted as comparative heatmaps for both groups. Heatmaps are based on mean VST-counts (see Materials and methods) and are sorted by peak expression times. Expression is normalized for both phenotypes together, meaning amplitudes are comparable between rhythmic and arrhythmic worms, but not between different transcripts. RAIN results are provided in S2E and S2F Table. Full lists of significant GO-terms is provided in S3A, S3B and S3C Table. VST-counts for heatmap generation are provided in S2C and S2D Table (not normalized).

terms occurred significantly more frequently in either behaviorally rhythmic or arrhythmic worms. The annotations of transcripts cycling in rhythmic, arrhythmic, or both behavioral phenotypes were compared against all GO-terms of the expressed 48,605 unique transcript IDs (S3 Table). This identified 77 GO-terms as enriched in the transcripts cycling in rhythmic worms, 29 in the transcripts cycling in arrhythmic worms, and 25 terms in the transcripts cycling in both phenotypes (Fig 3A–3D and S3 Table).

Cycling transcripts shared by worms of both phenotypes showed enrichment in GO-terms associated in rhythmic/circadian processes such as "circadian regulation of gene expression" and "entrainment of circadian clock" (Fig 3C), consistent with the described indistinguishable circadian clock transcript kinetics (Fig 2C–2L). There were further a number of GO-terms associated with light perception like "protein-chromophore linkage" and "cellular response to light stimulus." Transcripts significantly cycling only in rhythmic worms showed enrichment for a variety of GO-terms with a notable accumulation of terms associated with nervous system function and behavior (Fig 3B). Five of the top 10 most representative GO-terms were "neuromuscular process controlling balance," "regulation of long-term neuronal synaptic plasticity," "mechanosensory behavior," "axon regeneration," and "visual behavior." Similar terms were not enriched in transcripts cycling only in arrhythmic worms (Fig 3D). In behaviorally arrhythmic worms, the types of significantly cycling transcripts markedly differed from those of the rhythmic worms. They showed an enrichment for GO-terms associated with several metabolic and physiological pathways (Fig 3D). This highlights that behavioral rhythmicity and metabolic/physiological rhythmicity do not necessarily correlate and can in fact be highly contrasting.

In addition to these "neuronal/behavioral" versus "metabolic/physiological" terms, the GO-term "extracellular matrix disassembly" and others related to the extracellular space appeared prominently for both behavioral groups (Fig 3B–3D). A check of the associated transcripts against the NCBI nr-database showed that many encode for extracellular peptidases of the matrix metalloproteinase (MMP) family. Of the 8 putative MMP transcripts that showed 24-h cycling, 3 did so in rhythmic worms, 2 in arrhythmic worms, and 3 in both phenotypes. In *Drosophila*, MMPs are involved in the circadian remodeling of clock neuron axonal projections, together with *mef2* [45–47], which also cycles in our work. Given the differential cycling in behaviorally rhythmic versus arrhythmic worms, MMPs could be an interesting future target for closer investigation of circadian phenotype rhythm control.

The identification of distinct groups or GO-terms associated with either rhythmic or arrhythmic worms, while at the same time transcripts rhythmic for both behavioral phenotypes were enriched for core circadian clock components, strongly supports our interpretation that the distinctly cycling transcripts in behaviorally arrhythmic worms are not just due to random noise.

We next used the identified distinct GO-term groups to test whether the differences in transcript cycling were driven by reduced phase synchronization or absent/reduced cycling in one phenotype. A reduction of phase synchronization of transcript between different individual worms while individual worms maintained their amplitude (scenario A) should result in an overall increased variance between biological replicates (note: we depict standard deviation). If the transcripts indeed do not cycle in individual worms (scenario B), the variance should overall be less across the different biological replicates. The 2 scenarios were inspected for transcripts with significantly cycling only in rhythmic worms (GO-terms related to behavior or neuronal processes) or only in arrhythmic worms (GO-terms related to metabolism). Expression pattern variance analyses showed that indeed both scenarios were supported by individual transcripts. (S8C and S8D Fig). It is also possible that some of the higher variance could also result from random, non-diel transcript changes, which would also be in support of our interpretation. Together, this suggests that while in some cases indeed a lower phase synchronization might be the reason for the loss of transcript rhythmicity, in other cases, transcripts are clearly diel cycling in one behavioral group, but not the other (S8 Fig). This further strengthens the notion that behavioral arrhythmicity does not equal overall molecular arrhythmicity in the organism, even in the brain that has a strong impact on locomotor behavior.

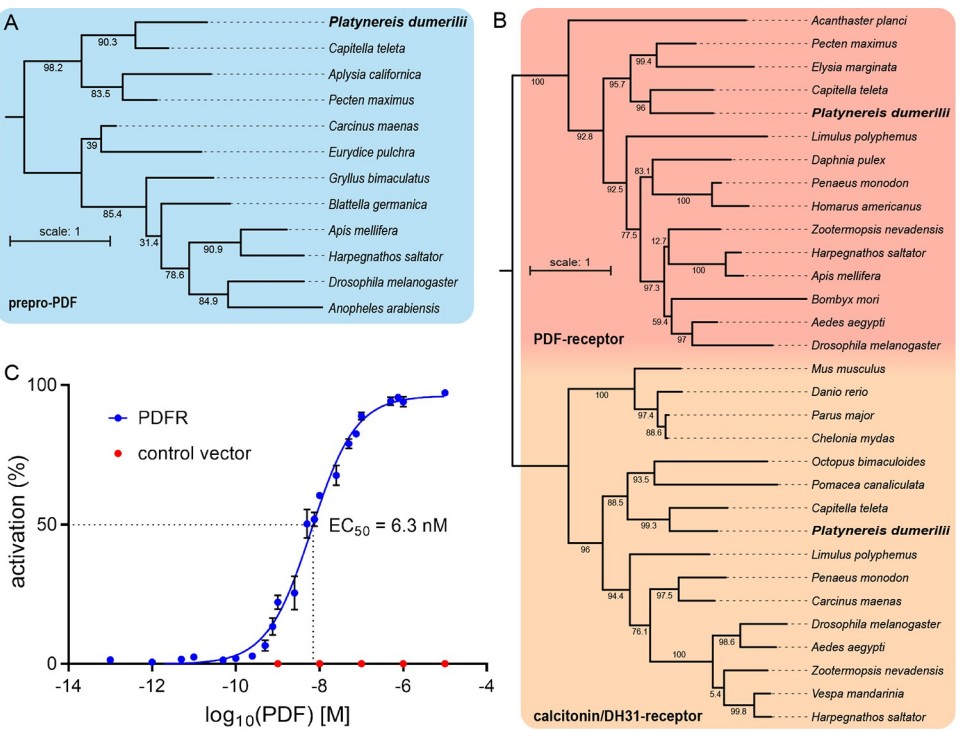

**Fig 4. PDF and PDF-receptor phylogeny and deorphanization.** Maximum-likelihood phylogenetic trees of (**A**) the pigment-dispersing factor precursor protein (prepro-PDF) and (**B**) the pigment-dispersing factor receptor (PDF-receptor), using the calcitonin-receptor and diuretic hormone 31 (DH31) receptor as outgroup. (**C**) Activation curve of the *P. dumerilii* PDF-receptor by PDF (blue) with $EC_{50}$ indicating the concentration needed for 50% receptor activation ($n = 10–12$ per concentration). Red dots indicate activation after transfection with empty control vector ($n = 6$ per concentration). Mean values ± SEM are shown. Peptide used sequences for phylogenetic tree construction are provided in S4A and S4B Table. Raw data of deorphanization measurements are provided in S5 Table.

## *Platynereis* possesses pigment-dispersing factor (PDF) and a functional PDF-receptor ortholog

In drosophilid flies and other insects, the conserved neuropeptide PDF is central to the communication and output of circadian clock neurons, thereby shaping circadian behavioral rhythmicity [16,48]. Given this connection between PDF and circadian behavioral control, we wondered about a possible role of PDF in *Platynereis* behavioral rhythmicity. We identified *Platynereis* sequences of the PDF precursor protein (prepro-PDF) and PDF-receptor (PDFR) and performed phylogenetic analyses (Fig 4A and 4B and S4 Table). This confirmed the identity of our PDF and PDFR sequences (Fig 4A and 4B and S4 Table).

To validate the functionality of the PDF/PDFR-system in *Platynereis*, *pdfr* was expressed in CHO cells employing an *apoaequorin/human Gα16 subunit* reporter system [49]. The assay confirmed that *Platynereis* PDFR can be activated by PDF at physiological concentrations ($EC_{50} = 6.3$ nM; Fig 4C and S5 Table).

Next, we induced mutations in the *pdf* gene of VIO wild-type worms using TALENs [50], which resulted in insertion/deletions in the third exon preceding the start of the mature peptide sequence (Fig 5A and 5B). Mutant worms were outcrossed against VIO wild types (at least 3 times). The resulting heterozygous offspring were in part incrossed to start the analyses of homozygous mutant worms, as well as further outcrossed. Thereby, we established 2 mutant allele lines with a −14-bp deletion and a +4-bp insertion, respectively (Fig 5A and 5B). Both alleles are predicted to result in a complete failure of mature peptide generation (Fig 5B),

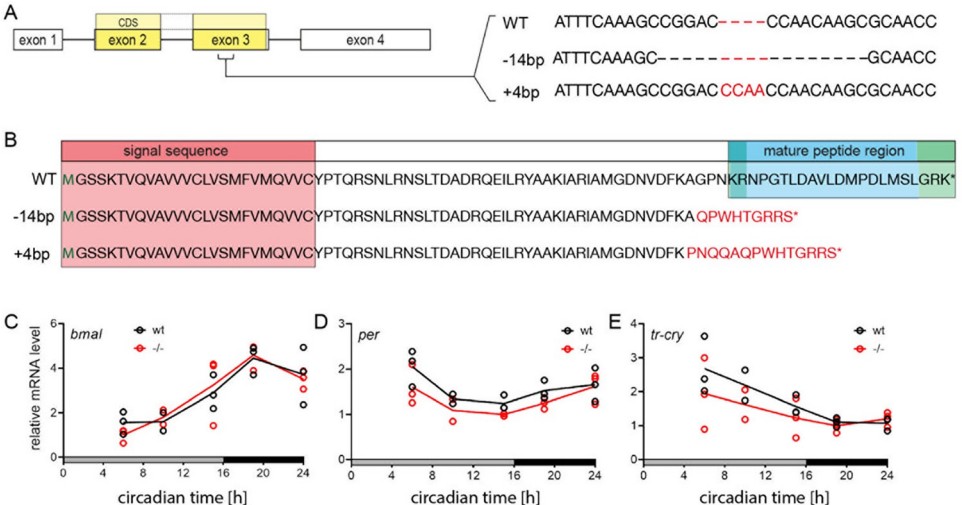

**Fig 5. TALENs-induced *pdf* gene knockout and effects on clock genes.** (**A**) TALENs targeting exon 3 of the *pdf* gene were used to induce frameshift mutations in the coding sequence (CDS). Two mutant alleles, a −14-bp deletion and a +4-bp insertion, were established. (**B**) Translated peptide sequence from the *pdf* wt and mutant loci. Start codon methionine (green M), signal sequence for cellular export (red background), cleavage regions (green background), and mature peptide (blue background) are indicated. Both frameshift mutations led to a premature stop codon (*) and a complete loss of the mature PDF peptide (see S9 Fig for confirmation by anti-PDF antibody). (**C-E**) Circadian expression of the clock genes *bmal*, *per*, and *tr-cry* in *pdf* wild types and mutants. Samples were collected on the second day of DD after being cultured under LD (16 h:8 h). Per time point, *n* = 3 replicates were measured (*n* = 2 for circadian time 10). Unpaired 2-sided *t* tests for each time points found no significant differences between wild types and mutants. Raw $C_t$-values of qPCR analyses are provided in S6 Table.

which was confirmed by immunohistochemistry with an antibody raised against the mature PDF peptide (S9 Fig).

   We determined the potential effects of *pdf* loss on the circadian clock by comparing the expression of the clock genes *bmal*, *per*, and *tr-cry* in *pdf* wild types and −14/−14 mutants from the initial VIO background (see next section and Materials and methods). Worms initially cultured under LD (16 h:8 h) were transferred to DD. Clock gene expression in worm heads was analyzed by qPCR on the second day of DD at 5 circadian time points. Unpaired 2-sided *t* tests found no significant differences between wild types and mutants for any gene and time point (Fig 5C–5E).

## *pdf* mutants show consistent increase in locomotor activity levels

To investigate the role of PDF in *Platynereis* circadian behavior, the activities of *pdf* wild types and mutants in a mixed VIO/PIN strain background were recorded for 4 d of LD (16 h:8 h) followed by 8 d of DD (Fig 6A). The behavioral characterization showed no systematic differences between mutant alleles (−14/−14, +4/+4), as well as transheterozygous worms (−14/+4), and thus data were combined (S10 Fig). *pdf* mutants consistently showed significantly elevated locomotor activity over the whole 24-h cycle both in LD and DD (Fig 6B). While the first mutants in the initial VIO background showed a small decrease in rhythmicity under LD and DD (S11 Fig and S1D Table), this decrease disappeared upon further outcrossing (Figs 6 and S12 and S1C and S1E Table). In fact, the rhythmicity trend was reversed in lines that had been further outcrossed, showing increased rhythmicity under LD, but not DD (Figs 6C, 6D, S12C and S12D). As this effect was consistently maintained in more outcrossed worms in different strain backgrounds (Figs 6 and S12), we interpret the initially observed decrease in rhythmicity (S11 Fig) rather as an unspecific background effect. We can, however, also not fully exclude

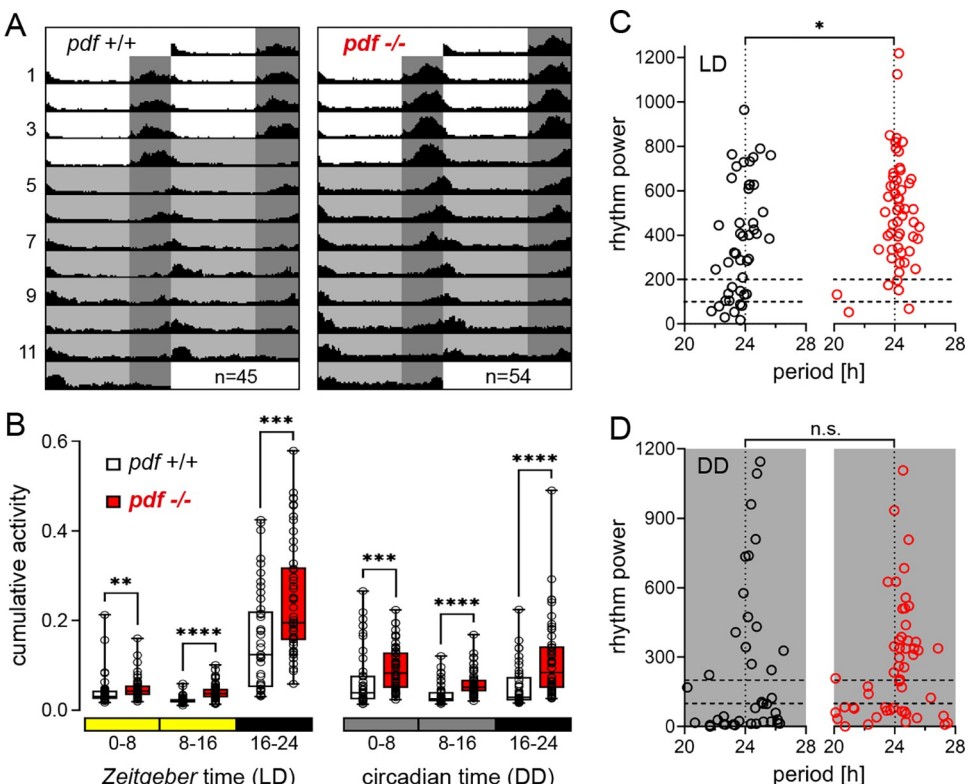

**Fig 6. *pdf* modulates overall locomotor activity, but not rhythmicity.** (**A**) Circadian locomotor activity of *pdf* wild types (black) and mutants (red) under 4 d of LD and 8 d of DD. Individual worm actograms are provided in S10A and S10B Fig. (**B**) Cumulative activity over the early day (0–8), late day (8–16), and night (16–24) in LD and DD. Box: median with 25%/75% percentiles, whiskers: min/max. (**C, D**) Period/power of wild type and mutant locomotor rhythms in the circadian range (20 h–28 h) in LD and DD determined by Lomb–Scargle periodogram. Statistical analyses (panels B–D) were performed via Mann–Whitney U-test. Significance levels: *$p < 0.05$, **$p < 0.01$, ***$p < 0.001$, ****$p < 0.0001$. Statistical comparisons of LD and DD phases in panel B are provided in S7 Table. Period/power values for panels C and D are provided in S1C Table.

that strain background might play an additional role. We believe that trusting the results from more outcrossed animals is a justified approach and, hence, refer to them further below.

The comparison of locomotor activity between LD and DD further showed that the *pdf* wild type tends to spread out its activity more across the 24-h cycle under DD (S7 Table), consistent with previous observations [37]. Under LD, part of the nocturnal worm's activity is thus likely suppressed by the acute light (masking). In the mutants, this might compress part of the overall increased activity into the available dark portion (S7 Table), providing a possible explanation for the consistently observed significantly higher rhythmicity of the mutants under LD, but not DD (Figs 6C, 6D, S12C and S12D and S1C and S1E Table).

Together, the results suggest that while PDF controls behavior in *Platynereis*, it is not a major determinant of circadian behavioral rhythmicity. Consistent with this, PDF immunohistochemistry (S9A–S9B' Fig) showed widespread expression in the *P. dumerilii* head and only a limited overlap with the oval-shaped domains between the posterior eyes that are characterized by high expression of core circadian clock genes [33,37].

## Discussion

Diversity of rhythmic phenotypes, even among closely related individuals, is ubiquitous in animals. However, especially for "nonconventional" but evolutionary and ecological informative

model systems investigations are still largely limited to the behavioral level [51–53]. Furthermore, it is largely unclear how much behavior can serve as an indicator of the overall rhythmicity of an individual and, in extent, the population. Here, we uncovered circadian behavioral patterns of the marine annelid worm *P. dumerilii*, which are individual specific and reproducible over time. Differences in the strength of behavioral rhythms did not correlate with transcript oscillations of core circadian clock genes, nor with the relative fractions of cycling transcripts in the transcriptome. Behavioral rhythm differences did, however, correlate with the types of transcripts showing cyclic expression: neuronal/behavioral versus metabolic/physiological transcripts correlating with behaviorally rhythmic versus arrhythmic worms, respectively.

It is noteworthy that worm behavioral phenotypes exist as a continuum from highly rhythmic to arrhythmic, with individual-specific pattern likely emerging from cumulative polygenic effects.

## *Platynereis* strains exhibit large and reproducible individual diversity of circadian behavioral rhythmicity

The individual diversity in worm diel phenotypes is remarkable, especially given that many of the investigated worms were siblings that were raised together in the same culture boxes. The reoccurrence of very similar behavioral patterns over time illustrates that these are not just behavioral snapshots of phenotypic plasticity but likely stable individual-specific characteristics. Similar reproducibility of individual behavior patterns has been described in several other species [29,54]. In a striking example, genetically identical freshwater fish were raised under near-identical conditions, but, nevertheless, individual-specific behavior patterns that persisted over several rounds of investigations were observable. This suggests that small environmental differences shaped those phenotypes, possibly via epigenetic modifications [55].

While our behavioral data suggest some strain-specific characteristics, there is still genetic variability within the strains, and, hence, those patterns may change further with continued incrossing due to the (random) selection for specific allele combinations. This might explain the behavioral differences observable between the VIO wild-type worms in the strain comparison (Figs 1A and S2C) and the VIO wild types in the *pdf* mutant analysis (Figs 6A and S10A).

## Transcript cycling patterns suggest contrasting rhythmicity strength in behavior versus physiology

In contrast to the divergent behavioral phenotypes, rhythmic or arrhythmic worms showed identical diel expression patterns of the core circadian clock genes. This shows that behavioral arrhythmicity was not the results of a disrupted or less synchronized circadian core clock. The few transcriptomic investigations of individual rhythmicity existing so far found correlations between diel activity pattern and different circadian clock components in forager versus nurse ants [14], as well as phenotype-specific expression of cryptochromes in zebrafish [54]. In *Drosophila*, a transcriptome comparison of early/late emerging fly chronotypes from different strains found no difference in core circadian clock genes [56], consistent with our observations that rhythmic behavioral control on the molecular level can be exerted downstream of the core circadian clock. Similarly, our results are in line with studies that found the same patterns of core circadian clock gene cycling in the suprachiasmatic nuclei of diurnal and nocturnal mammals [26,27], as well as in individual clock neuron groups of forager and nurse bees, which exhibit very different behavioral patterns across 24 h [57]. Notably, and different to our results, foragers and nurse bees did differ in circadian clock gene expression patterns when whole brains were analyzed.

Another aspect that could have affected transcript cycling is the exposure to LD cycles before/during sampling (masking). However, rhythmic and arrhythmic worms experienced the same conditions, meaning that the observed differences were due to a phenotype-specific regulation either downstream of the circadian clock or within direct light-response pathways. Thus, even if masking occurred, it was also phenotype specific. The fact that worms that are behaviorally clearly rhythmic or arrhythmic show more complex differences in gene expression cycling further emphasizes that rhythmicity monitored only via behavioral recordings or through a limited number of candidate genes (e.g., core circadian clock genes) is not necessarily representative of overall organismic rhythmicity. This is especially highlighted by the considerable number of transcripts cycling exclusively in behaviorally arrhythmic worms. As physiology can—much like behavior—be very much individual specific [58], our results suggest that chronobiological phenotype investigations should routinely consider multiple circadian clock outputs from different levels of biological organization.

Of note, a study under sensory temporal conflict conditions in the sea anemone *Nematostella vectensis* showed that a misalignment of environmental light and temperature cycles resulted in disruption of the circadian behavior [59]. However, despite the arrhythmic behaviors under sensory conflict conditions, the number of 24-h cycling transcripts was similar to the control conditions, with less than half overlapping. Furthermore, disruption of behavioral rhythms under sensory temporal conflict was primarily driven by diverging individual-specific responses rather than a general reduction of rhythmicity. Again, these findings very much align with our own results that show rhythm differences between levels of biological organization (behavior, circadian clock, gene expression), as well as individual-specific rhythmic phenotypes.

We hypothesize that the enriched GO-terms associated with transcripts only cycling in either rhythmic or arrhythmic worms reflect the differences in the respective lifestyles. For behaviorally rhythmic worms, nocturnal phases of activity are likely associated with foraging excursions where worms leave their tubes in search of food. The cycling of transcripts linked to behavioral processes as well as neuronal plasticity can ensure that at these times locomotor coordination and environmental responsiveness (e.g., for predator avoidance) is optimal. In contrast, the enrichment of metabolism/physiology-related GO-terms in behaviorally arrhythmic worms may be beneficial in habitats where food is readily available and major foraging excursions are not needed (e.g., the surface of a seagrass leaf). Under these conditions, worms can focus on the optimal processing of consumed energy, which can be fostered by temporal orchestration through the cyclic expression of metabolic genes. Similar patterns have been observed in Arctic vertebrates that show arrhythmic behavior during the midnight sun period in summer to maximize their food intake during the short productive period [25,60]. Importantly, these examples also show that clock oscillations and physiological rhythms can persist in times of arrhythmic behavior.

*Platynereis* live in heterogeneous coastal habitats and practices broadcast spawning where planktonic larvae get dispersed by ocean currents [31,32]. Under these conditions, high phenotypic diversity may ensure that at least part of the offspring are well adapted to the habitat they encounter upon benthic settlement, thereby carrying on their parent's genetic information (bet-hedging) [61,62]. On the individual level, what might be evolutionarily conserved is not a specific rhythmic phenotype per se, but rather the ability to produce offspring with a diversity of circadian rhythmic phenotypes. It is further important to note that in the heterogeneous environment inhabited by *P. dumerilii*, there are likely multiple phenotypes with "optimal" fitness depending on the individual microhabitat. Such coexistence of several different genotypes/phenotypes within the same population can allow for balancing evolution [63]. In fact, the persistence of multiple genotypes within a population seems to be especially common in

broadcast spawners inhabiting heterogeneous environments [64,65], and even for clock genes that are typically under strong selection for optimal timing and fitness, multiple alleles can persist in the same population [23,66,67]. While at present we cannot distinguish to which extent the diversity described here for *Platynereis* is caused by genetic versus environmental factors, evolutionary selection acts on the phenotype, meaning that the same principles apply.

The enhanced metabolic/physiological rhythmicity in behaviorally arrhythmic worms again illustrates that integrative investigations of organismal rhythmicity can be highly beneficial in revealing unexpected rhythmic complexity in any species. Ideally, such studies should also include proteomic investigations to produce an even more comprehensive picture of organismic rhythmicity [68].

## Conclusions

We identify consistent circadian behavioral rhythmic diversity among *P. dumerilii* strains and individuals, correlated with previously unknown complex molecular features.

Individual *Platynereis* worms can be discriminated based on their reproducible circadian locomotor patterns. The behavioral rhythmic phenotypes are independent of circadian clock transcript cycling, and, in contrast to *Drosophila* [69], loss-of-function mutations of the neuropeptide PDF have no effect on *Platynereis* circadian rhythmicity. Our results argue that circadian rhythmicity should not be characterized through behavioral phenotypes or indicator genes alone, but rather as the sum of various processes acting on different levels of organization. Most notably, our observation of enhanced diel cycling of metabolic/physiological transcripts in behaviorally arrhythmic worms highlights the complex relationship between the circadian clock and its behavioral and physiological outputs.

Investigating the factors that shape phenotypic and rhythmic diversity in an integrative way will help to understand how animals adapt to specific habitats and how these factors can contribute to overall population fitness. Differences between individuals and between levels of organization are common features of biological rhythm in various clades and habitats [59,70–72], and individual differences in behavioral patterns directly feed back to the experienced ("realized") environmental cycles used for entrainment [73–75]. It is thus important to treat biological variation not as "background noise," but as an additional aspect of biodiversity that needs to be considered in both the mechanistic and the ecological context [28,76].

## Materials and methods

### Experimental model and subject details

We performed all investigations on the marine annelid *P. dumerilii* (Audouin & Milne Edwards, 1833). The worms belonged to 3 wild-type strains, which all originate from worms collected in the Gulf of Naples, Italy (40.7° N, 14.2 E). The PIN and VIO strains have been in culture for approximately 70 years and have been incrossed at least 10 times [34,39]. The NAP strain was established in 2009 and has been kept as closed culture without targeted incrossing (Table 1). For all experiments, we used only immature/premature worms (age: 3 to 7 month, min. length: approximately 2.5 cm) that did not show any signs of metamorphic maturation to the adult spawning form (epitoke).

### Worm culture conditions

All worms used in the described experiments were cultured at the Max Perutz Labs (Vienna, Austria) in a 1:1 mix of natural seawater collected in the German bight by the Alfred-Wegener-Institute (Bremerhaven, Germany) and artificial seawater (Meersalz CLASSIC, Tropic

**Table 1. Resources (consumables, animals, cell lines, sequences, software) and deposited data.**

| Resource/Data | Source | Identifier |
|---|---|---|
| **Reagents, commercial assays, and other consumables** | | |
| OneTaq 2X Master Mix | New England BioLabs, USA | M0482S |
| RNAzol RT | Sigma-Aldrich, USA | R4533 |
| Direct-Zol RNA MiniPrep kit | Zymo Research, USA | R2052 |
| LunaScript RT SuperMix kit | New England BioLabs, USA | E3010S |
| Luna Universal qPCR Master Mix | New England BioLabs, USA | M3003S |
| GelGreen Nucleic Acid Gel Stain | Biotium, USA | 41005 |
| NEBNext Multiplex Oligos for Illumina | New England BioLabs, USA | E6440S |
| Meersalz CLASSIC sea salt | Tropic Marine, Switzerland | 10134 |
| Organic Spirulina powder | Micro Ingredients, USA | SPA-0520 |
| TetraMin fish flakes | Tetra GmbH, Germany | 769939 |
| Magnesiumchlorid hexahydrate ($MgCl_2$) | VWR International | 25.108.295 |
| Paraformaldehyde (PFA) | Sigma-Aldrich | 441244 |
| Methanol | VWR International | 20847.307P |
| TWEEN 20 | Sigma-Aldrich | P1379-500ML |
| Phosphate Buffered Saline (PBS), 10× | Sigma-Aldrich | P5493-1L |
| Proteinase K | VWR international | 1.24568.0100 |
| Glycin PUFFERAN, min. 99%, p.a. | Roth/Lactan | R3908.3 |
| Sheep Serum | VWR international | S2350-500 |
| Alexa Fluor 555 Azide, Triethylammonium Salt, 0.5 mg | Thermo Fisher Scientific | A20012 |
| Invitrogen Hoechst 33342, Trihydrochloride, Trihydrate—10 mg/mL Solution in Water | Thermo Fisher Scientific | H3570 |
| GLYCEROL, FOR MOLECULAR BIOLOGY, > = 99.0% | Sigma-Aldrich | G5516-1L |
| **Animal strains and cell lines** | | |
| *P. dumerilii* PIN strain | Zantke et al. 2014 [39] | N/A |
| *P. dumerilii* VIO strain | Zantke et al. 2014 [39] | N/A |
| *P. dumerilii* NAP strain | Zantke et al. 2014 [39] | N/A |
| *P. dumerilii* $pdf^{-14/-14}$ mutant strain | this paper | N/A |
| *P. dumerilii* $pdf^{+4/+4}$ mutant strain | this paper | N/A |
| Chinese hamster ovary (CHO) cells | PerkinElmer, Belgium | ES-000-A2 |
| *E. coli* competent cells (XL1-blue) | Tessmar-Raible group, Max Perutz Labs | N/A |
| **Primers, recombinant DNA, and proteins** | | |
| *pdf* genotyping primers<br>fwd: 5′- GCGCTATGCTGCAAAAATAGCTCGC-3′<br>rev: 5′- CGAGGGACATAAGGTCGGGCAT-3′ | this paper | N/A |
| *pdfr* cloning primers<br>fwd: 5′-GCTAAGCTTGCCACCATGCACGCCCTTCC-3′<br>rev: 5′-GGCTCTAGATCACCCCCCTGGACGTTGGTCAC-3′ | this paper | N/A |
| *bmal* qPCR primers<br>fwd: 5′- TCCGATTTATCTCCACGAGAA-3′<br>rev: 5′- TCCGTCTTTACAGGCAGCA-3′ | Veedin-Rajan et al. 2021 [32] | N/A |
| *per* qPCR primers<br>fwd: 5′-GGTCAACATGAAGTCGTACAGG-3′<br>rev: 5′- CACTGGTTTTCGGCTCAAG-3′ | Veedin-Rajan et al. 2021 [32] | N/A |
| *trcry* qPCR primers<br>fwd: 5′- TGTTCAAATTCCATGGGACA-3′<br>rev: 5′- TGTTTTAGCCTCAGCCCATT-3′ | Veedin-Rajan et al. 2021 [32] | N/A |
| *cdc5* qPCR primers<br>fwd: 5′- CCTATTGACATGGACGAAGATG-3′<br>rev: 5′- TTCCCTGTGTGTTCGCAAG-3′ | Veedin-Rajan et al. 2021 [32] | N/A |

(*Continued*)

**Table 1.** (Continued)

| Resource/Data | Source | Identifier |
|---|---|---|
| **Reagents, commercial assays, and other consumables** | | |
| pCDNA3.1(+) cloning vector | Thermo Fisher Scientific, USA | V79020 |
| pjet1.2 blunt vector (CloneJET PCR Cloning Kit) | Thermo Fisher Scientific, USA | K1231 |
| pGEM-T Easy Vector (pGEM-T Easy Vector Systems Kit) | Promega, USA | A1360 |
| Synthetic amidated PDF peptide for deorphanization (sequence: NH2-NPGTLDAVLDMPDLMSL-COOH) | this paper | N/A |
| *pdf* fwd TALEN (RVDs: NI-NI-NG-NN-NG-NN-NN-NI-NG-NG-NG-HD-NI-NI-NI-NN-HD) | this paper | N/A |
| *pdf* rev TALEN (RVDs: HD-HD-NI-NN-NG-NN-NG-NN-HD-HD-NI-NN-NN-NN-NG-NG-NN) | this paper | N/A |
| **Software** | | |
| RStudio v.1.1.453 | R Development Core Team 2013 [106] | https://www.r-project.org |
| R-package "DESeq2" v.1.30.1 | Love et al. 2014 [88] | https://bioconductor.org/packages/DESeq2/ |
| R-package "RAIN" v.1.24.0 | Thaben & Westermark 2014 [43] | https://bioconductor.org/packages/rain/ |
| R-package "topGO" v.2.46.0 | Alexa & Rahnenführer 2009 [95] | https://bioconductor.org/packages/topGO/ |
| R-package "gplot" v.3.1.1 | Warnes et al. 2016 [89] | https://CRAN.R-project.org/package=gplots |
| R-package "RColorBrewer" v.1.1–2 | Neuwirth 2014 [90] | https://CRAN.R-project.org/package=RColorBrewer |
| ImageJ/Fiji v.1.52d | Schindelin et al. 2012 [80] | https://imagej.nih.gov/ij/index.html |
| ActogramJ v.1.0 | Schmid et al. 2011 [81] | https://bene51.github.io/ActogramJ/ |
| StepOne v.2.3 Software for qPCR analysis | Thermo Fisher Scientific, USA | https://www.thermofisher.com |
| salmon v.1.4 | Patro et al. 2017 [87] | https://combine-lab.github.io/salmon/ |
| SAMtools v.1.3.1 | Li et al. 2009 [107] | http://samtools.sourceforge.net |
| cutadapt v.1.12 | Martin 2011 [84] | https://cutadapt.readthedocs.io/en/stable/index.html |
| FastQC v. 0.11.9 | Andrews 2010 [85] | http://www.bioinformatics.babraham.ac.uk/projects/fastqc/ |
| GraphPad Prism v.9.2.0 | GraphPad Software, USA | https://www.graphpad.com |
| Adobe Illustrator v.25.3.1 | Adobe, USA | https://www.adobe.com/at/products/illustrator.html |
| TransDecoder | Haas et al. 2013 [108] | https://github.com/TransDecoder/TransDecoder |
| HMMER v.3.3.1 | Mistry et al. 2021 [94] | http://hmmer.org |
| **Deposited data** | | |
| *Platynereis* tail reference transcriptome for mapping | this paper | Dryad doi: 10.5061/dryad.31zcrjdnq |
| .fastq-files with trimmed forward and reverse reads (2 files/sample) | this paper | |
| .fastqc-files with quality reports for trimmed.fastq-files | this paper | |
| Python code for binary behavior reproducibility analysis | this paper | |
| R-script for GO-term analysis incl. associated files | this paper | |

Marine AG, Switzerland). The water was pumped through 1 μm filters before use and was exchanged every 1 to 2 weeks. Worms were kept in transparent plastic boxes (30 × 20 cm, 100 to 150 indv./box) in temperature controlled rooms at 19.5 ± 1.0°C under a rectangular 16 h:8 h

LD with fluorescent tubes ($4.78 * 10^{14}$ photons $* cm^{-2} * s^{-1}$, Master TL5 HO 54W/865, Philips, the Netherlands, S1A and S1C Fig). To account for *Platynereis*' lunar reproductive cycle [37,77,78], we simulated a moonlight cycle by an LED lamp (Warm White 2700K A60 9W, Philips, the Netherlands) that was switched on for 8 nights centered on the outdoor full moon times (see [38] for details). Larvae worms were fed a mix of living *Tetraselmis* algae and *Spirulina* (Organic Spirulina Powder, Micro Ingredients, USA) 2 to 3 times a week until they started forming their living tubes. Thereafter, worms were fed shredded spinach and powdered fish flakes (TetraMin, Tetra GmbH, Germany) (once a week, each).

## Recording and analysis of worm locomotor behavior

We investigated worm locomotor activity in light-tight chambers developed in cooperation with loopbio GmbH, Austria [32,79]. The worms were last fed (spinach) 4 d before the start of the investigation. Worms were always taken from culture boxes of similar animal densities and age to avoid housing effect on behavior. All worms were inspected for signs of sickness (twitching, no tube-building, changed color, inclusions/ulcers, injuries) and only healthy-looking worms were used. To prevent potential behavioral changes associated with the simulated lunar cycle (see above), we conducted all recordings in the 2 weeks around new moon. For recording, the worms were transferred to $5 \times 5$ well plates and were filmed at 15 fps using infrared-illumination and a camera equipped with a long-pass IR-filter (acA2040-90 um, Basler AG, Germany). A custom-build LED-array was used to create a 16 h:8 h LD cycle with natural spectral composition ($1.40 * 10^{15}$ photons $* cm^{-2} * s^{-1}$, Marine Breeding Systems GmbH, Switzerland, S1B and S1C Fig). All recordings were done at culture temperature, and diel oscillations were approximately 1˚C in LD and <0.5˚C in DD. For data analysis, we excluded worms that started maturing during or within 1 week after the recording, as worm behavior is strongly altered during the maturation phase. Data of worms that crawled out of the well and could no longer be recorded were omitted, if less than 3 d of data were available for the respective recoding phase (LD or DD). Sex determination in a limited number of wild-type worms showed no sex-specific differences in locomotor behavior (S2 Fig), and, hence, data of males and females were pooled.

We quantified locomotor activity by determining x/y-coordinates of the center of the detected worm shape (Fig 2A) and measuring the distance moved between frames with the Motif automated tracking software (loopbio GmbH, Austria; see [32] for details). We used a 1-min moving average to reduce noise. Period and power of diel rhythmicity was determined via Lomb–Scargle periodograms with the ActogramJ plug-in in ImageJ/Fiji [80,81]. Analysis focused on the 20-h to 28-h range of circadian rhythmicity. Overall activity was binned into morning (ZT0-8), afternoon (ZT8-16), and night (ZT16-24) for the LD phase, and into subjective morning (circadian time, CT0-8), subjective afternoon (CT8-16), and subjective night (CT16-24) for the DD phase with all days of the respective phases being pooled.

We compared circadian rhythmicity power between groups, as well as overall activity between time bins (morning, afternoon, night) and between groups via unpaired 2-sided *t* test or via one-way ANOVA with Holm–Sidak post hoc test. If data were not normally distributed and/or homoscedasticity of variances was not given, we replaced the *t* test with a Mann–Whitney U-test and the one-way ANOVA with a Welch's ANOVA with Dunnett-T3 post hoc test or a Kruskal–Wallis ANOVA on ranks with Dunn's post hoc test. Statistical analyses were carried out in GraphPad Prism (GraphPad Software, USA).

The number of recording days in LD or DD depended on the specific investigation. For the comparison of the PIN, NAP, and VIO wild-type strains (Fig 1), we exposed the worms to 4 d of LD and DD, respectively. We recorded $n = 25$ PIN worms, $n = 21$ NAP worms, and $n = 38$ VIO worms (Figs 1 and S2). PIN, NAP, and VIO worms for the comparison originated from 2

mating batches, each. Repetition runs were done with 12 PIN worms, 5 NAP worms, and 13 VIO worms (Figs 1E–1G and S4), and the respective worms were kept separately in smaller boxes after the initial recording to avoid the confusion of individuals.

To behaviorally characterize worms for RNASeq analysis (Fig 2, described below), we exposed them to 3 d of LD and DD, respectively. Over the course of 6 mo, we performed 13 behavioral recordings using a total of 325 PIN wild-type worms from 5 different mating batches.

Worms in the comparison of *pdf* mutants and wild types (Fig 6) had a mixed VIO/PIN background and experienced 4 d of LD followed by 4, 5, or 8 d of DD. The fractions of worms experiencing a given number of DD days did not differ between wild types and mutants, meaning that datasets are directly comparable. We performed 5 recordings using $n = 45$ *pdf* wild types from 5 mating batches and $n = 54$ *pdf* mutants from 6 mating batches ($-14/-14$ $n = 16$, $+4/+4$ $n = 31$, $-14/+4$ $n = 7$). As the mutants showed no within-strain differences among genotypes (S10 Fig and S1C Table), all mutant genotypes were pooled. Details on the recordings of *pdf* worms in the initial VIO strain background (S11 Fig and S1D Table) and after later being backcrossed into the VIO strain (S12 Fig and S1E Table) are provided in the respective Supporting information. The *pdf* genotypes of all mutants and the associated wild-type worms were verified at least 1 mo before the behavior recording or directly afterwards (described below).

Actograms of individual worms used in strain comparison, characterization for RNASeq, and *pdf* wild type/mutant comparison are provided in the Supporting information (S2, S6 and S10 Figs).

## Similarity analyses of individual worm behavioral patterns

To quantify the reproducibility of worm behavior in the initial versus the repeated recording of locomotor activity (Figs 1E and S4), we performed 2 independent statistical analyses. For the first approach, we employed an approach based on binary activity determination that found higher behavioral similarity for matching repetition runs in both LD and DD (see Results; S5 Fig).

For the second approach, we treated LD and DD data separately, and for each phase, we pooled the 4 d of data to one 24-h period with 30-min bins. Using the Kolmogorov–Smirnov test (KS-test) as a tool to quantify similarity between diel behavioral patterns [40], we separately compared each initial behavioral run to all repetition runs ($n = 30$). Resulting KS-test *p*-values are measures of the probability of dissimilarity between 2 curves (i.e., diel behavioral patterns). The lower the *p*-value, the less dissimilar (in other words, more similar) the data. We therefore treated KS-test *p*-values as proxies of behavioral similarity between initial and repeated runs. As the data structure was identical for all initial/repetition-comparisons (same number of LD/DD days for all runs), a direct comparison of KS-test *p*-values is possible. KS-test *p*-values were $\log_{10}$-transformed for visualization purposes on the y-axis. Next, we compared the *p*-values of each matching initial/repetition KS-test and those from the control group (i.e., the behavioral patterns of all other worms under the respective condition). This was done via one-sample Wilcoxon signed-rank test for each initial run in LD and DD. The resulting (Wilcoxon test) *p*-values were corrected for multiple testing via the *p.adjust()* function in R according to the Benjamini–Hochberg procedure [82] with *p*-values $< 0.05$ considered significant (Fig 1E, scatter plots).

## Rhythmicity characterization of wildtype worms and sampling for RNASeq analysis

PIN wild-type worms whose locomotor activity had previously been recorded (3 d LD and DD; see above) were first grouped based on their behavioral rhythmicity. We characterized

worms as rhythmic (*n* = 103) or arrhythmic (*n* = 74) based on Lomb–Scargle periodogram analysis in ActogramJ and visual inspection of actograms (S6 Fig). Worms were characterized as rhythmic if rhythm power in the circadian range (20 h to 28 h) was >150 in LD and/or DD, and worms had clearly visible nocturnal activity peaks in both LD and DD in the actograms (see S1B Table for period/power values). As periodogram analysis can struggle to recognize rhythmicity in the case of very short (but clearly rhythmic) activity peaks, a few additional worms with lower rhythm power were also characterized as rhythmic (S6 Fig: rec#4/worm#8, rec#4/worm#13, rec#7/worm#13, rec#9/worm#13, rec#10/worm#6, rec#10/worm#13, and rec#12/worm#6). Worms were characterized as arrhythmic if rhythm power in the circadian range was <150 in both LD and DD and worms showed no visible diel/circadian rhythmicity in the actograms. In some cases, periodogram analysis yielded higher rhythm power in LD that was, however, not at all supported by the actograms (S6 Fig: rec#1/worm#4, rec#1/worm#10, rec#2/worm#2, rec#2/worm#11, rec#4/worm#23, and rec#8/worm#15), and the respective worms were also characterized as arrhythmic. Importantly, even if individual worms had been mischaracterized (for which we have no evidence), these individuals would in the worst case blur the patterns identified via RNAseq but would not lead to any false patterns (artifacts). Worms that were neither clearly rhythmic nor arrhythmic were considered "intermediate" and were not used further (*n* = 125). Furthermore, *n* = 23 worms were excluded from analysis, because they matured or crawled out of the tracking well. While the relative abundance of phenotypes did partially differ between recordings, all included rhythmic and arrhythmic worms. Actograms of all individual worms including their rhythmic characterizations are provided in S6 Fig.

To further ensure that the supposed inactivity of arrhythmic worms was not related to any sickness, we replotted the respective actograms with 10-fold magnification of the y-axis (S7A Fig). This illustrates that activity was present and the comparison with worms that crawled out of the tracking well (S7B and S7C Fig) confirms that this is not "background noise," but actual worm movement. The reoccurring activity spikes visible in the magnification can further be attributed to normal worm behavior (moving/turning in tube, respiratory undulations) [37,83], and activity of arrhythmic worms in general is highly similar to that of rhythmic worms during (subjective) daytime in both LD and DD (S6 Fig). Additionally, as some worms matured between the behavioral characterization and the sampling for RNAseq (see below), we could see that rhythmic and arrhythmic worms did this in exactly the same way and with the same reproductive success. Hence, there is no indication of arrhythmic worms being in any way compromised.

We sorted characterized worms to culture boxes based on rhythmicity phenotype and kept them under standard conditions (LD 16 h:8 h + lunar cycle) until sampling at the next new moon (i.e., approximately 1 mo after the behavioral characterization). Samples for RNASeq analysis were collected at ZT0 (lights on), ZT4, ZT8, ZT16 (lights off), and ZT20 (Fig 2A). Due to the work-intensive characterization procedure and the loss of characterized worms due to maturation, we could not collect all samples for the diel time series "in one go," but sampled discontinuously over several months while making sure that replicates of the same rhythmic phenotype/time point were not collected simultaneously. Worm age at the time of sampling was 4 to 7 mo. We last fed the worms 5 d before sampling (spinach), and 2 d before sampling, we transferred them to sterile filtered seawater containing 0.063 mg/mL streptomycin-sulfate and 0.250 mg/mL ampicillin to minimize bacterial contaminations. Worm heads were sampled by anesthetizing animals in 7.5% (w/v) $MgCl_2$ mixed 1:1 with sterile filtered seawater for 5 min. Heads were cut before the first pair of parapodia under a dissection microscope before being fixed in liquid nitrogen with 3 heads pooled per replicate. Dark samplings at ZT16 and TZ20 were conducted under dim red light. For each time point, we collected 3 replicates of

rhythmic and arrhythmic worms, respectively. During sampling, worms were also visually inspected for differences between rhythmic phenotypes (body size/shape/coloration, parasites), but none were found. RNA was extracted as described below (*pdf* mutant clock gene qPCR) and was stored at −80°C.

### Processing of RNASeq data

Quality control, reverse transcription, and library preparation of rhythmic/arrhythmic samples were performed by the VBCF Next-Generation Sequencing facility services according to standard procedures (VBCF-NGS, Vienna Biocenter Campus, https://www.viennabiocenter.org/vbcf/next-generation-sequencing). Samples were run on a NextSeq550 System (Illumina, USA) with the High Output-kit as 75 bp paired-end reads and resulted in a total of 445,553,522 raw reads (sample range: 9,973,579 to 15,234,688 reads).

We trimmed/filtered the raw reads from NextSeq sequencing using cutadapt (—nextseq-trim = 20 -m 35—pair-filter = any) removing adapters in the process [84]. This reduced read numbers to 443,393,107 reads (0.48% loss). Read quality was checked before and after trimming using FastQC [85]. Trimmed reads were mapped against a *Platynereis* reference transcriptome generated from trunk and regenerated tail pieces (Fig 2B). Trimmed-read fastq-files and FastQC-reports as well as transcript sequences of the reference transcriptome were deposited at the Dryad online repository (doi:10.5061/dryad.31zcrjdnq). We chose this tail transcriptome, because a BUSCO analysis [86] showed that it contained a higher percentage of near-universal single-copy orthologs (98.9% complete) compared to a previously published transcriptome generated from worm heads [41]. To ensure that no important transcripts were lost due to tissue-dependent expression, we manually verified the inclusion of known clock genes, opsins, neuropeptides, and other potential genes of interest. The respective transcripts were identified using internal sequence resources and NCBI and were verified via BLASTx against the nr-database. If sequences from existing *Platynereis* resources covered larger gene stretches than the respective transcript, we replaced the sequences in the transcriptome or merged them. Trimmed reads were mapped to the reference transcriptome using salmon tool [87] (standard parameters incl. multi-mapping, library type: ISR). To exclude transcripts that did not receive any reads during mapping, we removed those transcripts from the expression matrix that had no observed expression in at least 2 time points in both the rhythmic samples and the arrhythmic samples. We further excluded all transcripts <500 bp to avoid fragments that could not be reliably annotated, thereby obtaining a total of 48,605 expressed transcripts for differential expression by sequencing (DEseq) analyses (Fig 2B and S2 Table).

### Identification of cycling transcripts with RAIN

To identify cyclic changes in transcript expression patterns, we used the R-package "RAIN," which employs a nonparametric approach to identify cycling independent of waveform [43]. As RAIN assumes homoscedasticity, raw count data were transformed via the *VarianceStabilizingTransformation* function of the DESeq2 package and resulting VST-counts were analyzed [88]. Cycling was investigated separately for rhythmic and arrhythmic samples for a period length of 24 h with RAIN standard settings (peak.border = c(0.3,0.7)). We could not test for other periods due to the discontinuous sample collection, which would have disrupted non-24-h cycles (see above). Resulting *p*-values were false discovery rate corrected via the *p.adjust()* function in R according to the Benjamini–Hochberg procedure [82] with *p*-values < 0.05 considered significant (Fig 3A and S2 Table).

For visualization, we normalized VST-counts of cycling transcripts to a scale of −1 to 1 and plotted them in heatmaps using the R-packages *gplots* [89] and *RColorBrewer* [90]. Transcripts

with cycling in rhythmic worms, arrhythmic worms, or both were plotted separately, but always for both phenotypes (Fig 3B–3D). The normalization means that expression amplitudes of a given transcript can be compared across rhythmic and arrhythmic worms, but not between different transcripts or groups (rhythmic, arrhythmic, both). All used software versions are detailed in Table 1.

## GO-term annotation and enrichment analysis

We next performed GO analysis to determine the enrichment of biological processes among cycling transcripts. For this, transcripts were assigned GO-terms using the Trinotate protocol [91]. Briefly, we used TransDecoder to identify the most likely longest open reading frame (ORF) peptide candidates and performed a BLASTx or BLASTp search of the transcripts or peptide candidates against the SwissProt database (UniRef90) [92,93]. Additionally, we used the *hmmscan* function from HMMER to get functional information from protein domains by searching the Pfam database [94]. For analysis, we pooled the GO-terms hits that had e-values $<1 * 10^{-3}$ in any of the annotation approaches. GO-terms could be assigned to 19,096 out of the 48,605 expressed transcripts, and to 334, 273 and 176 of the transcripts cycling in rhythmic, arrhythmic, and both phenotypes, respectively.

We determined GO-term enrichment for cycling transcripts with the R-package "topGO" [95], using Fisher's exact test and the "elim" algorithm [95,96]. The elim algorithm avoids redundant scoring of more general GO-terms and thereby gives higher significance to more specific terms than the "classic" algorithm. GO-terms were considered significantly enriched, if they had a $p$-value $\leq 0.05$ and were represented by at least 3 transcripts (Fig 3B–3D and S3 Table). All used software versions are detailed in Table 1.

## Description of *Platynereis* pigment-dispersing factor (*pdf*) and receptor (*pdfr*) sequences

An initial *Platynereis pdf* gene fragment was identified from Expressed Sequence Tags (ESTs) [97]. The resulting 567 bp fragment was subsequently extended to 804 bp, which included the complete coding sequence (CDS) of the prepro-peptide (accession no˚: GU322426.1).

A first fragment of the *Platynereis* PDF-receptor (*pdfr*) sequence was obtained by degenerated PCR and subsequently extended by RACE-PCR [98]. A 1,604-bp sequence containing the full CDS was cloned into a pGEM-T Easy Vector (Promega, USA) and sequenced (accession no˚: OL606759). A draft genome of *P. dumerilii* (Genbank ID: GCA_026936325.1) was searched for possible gene duplicates of *pdf* using a hidden Markov model, but no further copies were identified. For the construction of phylogenetic trees for prepro-PDF and PDF-receptor (Fig 4A and 4B), we collected sequences from NCBI via PSI-BLAST (3 iterations), aligned them with MUSCLE [99] and then constructed the trees using the IQ-TREE online tool with default settings [100], and visualized via iTOL [101]. Sequences used for tree construction are provided in the supplement (S4 Table).

## Deorphanization of pigment-dispersing factor receptor

To validate that the identified *pdfr* sequence encodes a functional receptor activated by PDF, a fragment of the *pdfr* gene including the full CDS was amplified by PCR and cloned into the HindIII/XbaI sites of a pCDNA3.1(+) cloning vector. We assessed receptor activation as previously described [49]. Briefly, *pdfr*-transfected Chinese Hamster Ovary (CHO) cells expressing *apoaequorin* and the *human Gα16 subunit* were incubated with $10^{-13}$ to $10^{-4}$ M of synthetic amidated PDF with $n$ = 10 to 12 measurements per concentration. PDF was provided by the Mass Spectrometry Facility of the Institute for Molecular Pathology, Vienna, and assessed by

the Chemistry Department of the University of Vienna for purity using matrix-assisted laser desorption/ionization time-of-flight mass spectrometry (MALDI-TOF-MS). We recorded calcium responses for 30 s using a Mithras LB 940 luminometer (Berthold Technologies, Germany). After these ligand-stimulated calcium measurements, Triton X-100 (0.1%) was added to each sample well to obtain a measure of the maximum calcium response and to calculate dose responses. Percentage activation values were plotted, with cells transfected with an empty pCDNA3.1(+) vector serving as negative controls with $n = 6$ per concentration (Fig 4C and S5 Table).

The half-maximal effective concentration (EC$_{50}$) was calculated using a computerized nonlinear regression analysis with a sigmoidal dose–response equation in GraphPad Prism.

## Generation of *pdf*-mutant strain using TALENs

To achieve a knockout of the *Platynereis pdf* gene, we followed a reverse genetic approach using TALENs (transcription activator-like effector nucleases) as previously described [50]. Repeat-variable di-residue (RVD) sequence of TALENs were designed based on the previously identified *pdf* sequence (accession no˚: GU322426.1) to target a site in the third exon preceding the start of the mature peptide sequence (Table 1). VIO strain worm zygotes 1 h postfertilization were injected with 10 μL injection solution, which consisted of 200 ng/μL of each TALEN, 8 μL RNAse-free H$_2$O, and 2 μL TRITC-dextrane (3% in 0.2 M KCl). Before injection, the solution was filtered through 0.45 mm PVDF centrifugal filters (Ultrafree-MC-HV, Merck Millipore, USA) [50].

Once grown to maturity, we outcrossed injected worms against VIO wild-type worms, and the resulting offspring were genotyped to identify changes in the PCR-product length of the *pdf* locus using the HotSHOT-method [102] and primers covering the TALENs cleavage site. PCR products were run on agarose gels [103]. When changes in length compared to the wild type were found, the bands were cut out and subcloned according to manufacturer protocols into either a pjet1.2 blunt vector (CloneJET PCR Cloning Kit, Thermo Fisher Scientific, USA) or a pGEM-T Easy Vector (pGEM-T Easy Vector Systems Kit, Promega, USA), depending on the Taq-polymerase used for PCR (blunt versus sticky ends). The vectors were used to transform competent *E. coli* cells (XL1-blue, raised by lab technician), of which individual colonies were picked. Success of the transformation was verified by PCR followed by sequencing of the insert to identify base deletions/insertions [103]. We thereby established a −14 bp deletion mutant strain, a +4bp insertion mutant strain, as well as a corresponding wild-type strain (Fig 5A). The mutations resulted in early stop codons and a complete loss of the mature peptide sequence (Fig 5B). The mutants were initially crossed against VIO wild types at least 3 times before later also being outcrossed against PIN wild types to create a mixed VIO/PIN background. A rhythmic phenotype (reduced rhythmicity) observed in initial VIO strain *pdf* −14/−14 and −14/+4 mutants (S11 Fig and S1D Table) resembled *pdf* mutants phenotypes in *Drosophila* [16]. However, the phenotype disappeared after the outcross against the PIN strain (Fig 6 and S1C Table) and could not be recovered by backcrossing against VIO worms (S12 Fig and S1E Table), thus strongly indicating an off-target effect in the initial VIO mutants that was unrelated to the knockout of *pdf*.

## Generation of anti-*Pdu*-PDF antibody

Polyclonal antibodies against the amidated C-terminus (i.e., mature peptide) of *Platynereis* PDF (NPGTLDAVLDMPDLMSL-amid) conjugated to the carrier protein Ovalbumin were developed by the company PRIMM (Italy). Two different rabbits were immunized and the

serum immunoaffinity purified. The polyclonal AB from rabbit 1 provided the better results and was hence further used.

## Confirmation of PDF absence in mutants via immunohistochemistry (IHC)

Staining of PDF in immature *P. dumerilii* heads (S9 Fig) was done according to an established protocol [33]. Briefly, worm heads were cut and fixed in 4% PFA for 2 h at room temperature. Dehydration and permeabilization of samples was done using increasing concentrations of methanol (MeOH), with final storage of the samples in 100% MeOH (at least overnight). Proceeding with the immunohistochemistry protocol, samples were rehydrated in 1X PTW (1X PBS/ 0.1% TWEEN20) and digested for 6 min using Proteinase K (100 μg/mL), then rinsed 2× 2 min in glycine (2 mg/mL). Samples were next washed 5× (1-2-5-10-10 min) in 1X PTW. To improve visualization, tissue clearing and depigmentation was done using DEEP-Clear [104], for which samples were incubated in clearing solution for 20 min at room temperature, shaking. Heads were washed 5× 10 min in 1X PTW, then blocked overnight in 10% sheep serum (VWR International, USA) at 4°C. The following day, samples were incubated in anti-*Pdu*-PDF (1:125 in 5% sheep serum). Incubation in secondary antibody, Alexa Flour 555 goat anti-rabbit (1:400 in 1X PTW, Thermo Fisher Scientific, USA), followed. All antibody incubations were done for 3 d at 4°C while shaking and were followed by washing of the samples in 1X PTW for at least 4× 15 min at room temperature plus overnight at 4°C. Then, staining of nuclei was done using Hoechst 33342 (1:1,000 in 1X PTW, Thermo Fisher Scientific, USA). Samples were incubated at least overnight, followed by 3× 10 min washes in 1X PTW and equilibration of samples at least overnight at 4°C in 35% glycerol. Finally, the heads were mounted in 70% glycerol (Sigma-Aldrich, USA).

Imaging was done using a Zeiss LSM700 inverse confocal microscope with LD LCI Plan-Apochromat 25×/0.8 Imm Korr DIC M27 lenses (oil immersion). We performed Z-stack imaging with microscope parameters set for the specific staining in *pdf* +/+ worms and applied it to all samples. Using ImageJ/Fiji [80], Z-projections of average intensity were prepared for all samples. Min/Max range for the PDF channel (Alexa Fluor 555) was 0–343.

## Clock gene qPCR measurements in *pdf* mutant worms

After entrainment to a 16 h:8 h LD cycle, we sampled worm heads of *pdf* wild types and −14/−14 mutants (VIO background) on the second day of DD at circadian time point 6 (CT6), CT10, CT15, CT19, and CT24 as described above for RNASeq analysis. Animals were only briefly exposed to light during dissection. At each time point, 3 replicates of mutant and wild-type worms were collected with 5 heads pooled per replicate (only 2 replicates for CT10).

For RNA extraction, we homogenized samples in 350 μL RNAzol (Sigma-Aldrich, USA) with a TissueLyser (Qiagen, the Netherlands) for 2 min at 30 Hz and extracted RNA using the Direct-Zol RNA MiniPrep kit (Zymo Research, USA) with on-column DNA digest according to the manufacturer protocol. RNA was eluded in 25 μL Nuclease-free H$_2$O and was stored at −80°C. We spectrometrically determined RNA concentration and purity (Nanodrop 2000, Thermo Fisher Scientific, USA). Approximately 1 μg RNA was reverse transcribed to cDNA (LunaScript RT SuperMix, New England BioLabs, USA).

We measured the expression of the circadian clock genes *brain and muscle Arnt-like protein* (*bmal*), *period* (*per*), and *tr-cryptochrome* (*tr-cry* aka *mammalian-type cry*) via real-time qPCR (StepOnePlus, Applied Biosystems, USA) using previously established primers [32] and *cell division cycle 5* (*cdc5*) acting as a reference for normalization [41] (Fig 5C–5E and S6 Table). All biological replicates were also measured as technical duplicates.

We normalized raw $C_t$-values of the clock genes *bmal*, *per*, and *tr-cry* according to the $2^{-\Delta\Delta CT}$ method [105] using the gene *cdc5* as a reference (Fig 5C–5E). Expression stability of *cdc5* has previously been verified [41]. Raw $C_t$-values are provided in the Supporting information (S6 Table). Differences between worm groups were analyzed via unpaired 2-sided *t* tests for each time point and with correction for the testing of multiple time points. All statistical tests were performed using GraphPad Prism.

## Dryad DOI

Dryaddoi:10.5061/dryad.31zcrjdnq [109].

## Supporting information

**S1 Fig. Spectral light conditions for worm incubations.** (**A**) Standard worm culture light spectrum. (**B**) Behavior chamber light spectrum. (**C**) Logarithmic plotting of panels A and B as well as a natural sunlight spectrum recorded in the natural habitat of *Platynereis dumerilii* around Ischia, Italy, at 5 m depth in November 2011 (10 AM–4 PM local time average) [38]. Overall irradiance (380–750 nm) was $4.78 * 10^{14}$ photons $* cm^{-2} * s^{-1}$ for the worm culture and $1.40 * 10^{15}$ photons $* cm^{-2} * s^{-1}$ for the behavior chamber.
(JPG)

**S2 Fig. Individual worm actograms of strain comparison.** Related to Fig 1. Double-plotted actograms of individual worms from the (**A**) PIN strain, (**B**) NAP strain, and (**C**) VIO strain are shown. Locomotor activity was recorded over 4 d of LD (16 h:8 h) and 8 d of DD. #: individual worm identifier. Red shading indicates that worms crawled out of the tracking well. Worms that matured during or within 1 week after the recording were excluded from statistics and are not shown, as maturation strongly alters their overall behavior. For worms that matured later (used in statistics), sex is indicated, if known (not recorded systematically). Sexes were determined to check for potential sex-specific behavioral differences, of which none were found.
(PDF)

**S3 Fig. Magnified individual actograms of arrhythmic worms in strain comparison.** Related to Figs 1, S2 and S4. Double-plotted actograms of individual (**A**) PIN, (**B**) NAP, and (**C**) VIO wild-type worms are shown. Locomotor activity was recorded over 4 d of LD (16 h:8 h) and 4 d of DD. Y-axis is magnified 10-fold relative to S2 Fig to better visualization activity patterns. The comparison illustrates that although arrhythmic worms showed overall lower activity, they were far from inactive. Individual worm identifiers (#) match those in Figs S2 and S4.
(JPG)

**S4 Fig. NAP and VIO worm locomotor activity in initial and repetition runs.** Related to Fig 1. Double-plotted actograms of (**A**) NAP strain and (**B**) VIO strain individual worms are shown. Locomotor behavior was recorded over 4 d of LD (16 h:8 h) and 8 d of DD in 2 consecutive runs (initial/repetition). #: individual worm identifier. Red shading indicates that worms crawled out of the tracking well. Worms that were excluded from statistics due to maturation during or within 1 week after the recording are not shown, as maturation strongly alters their overall behavior. For an explanation of scatter plots on behavioral similarity in LD and DD (gray background), see Fig 1E. In the VIO strain (**B**), 7/13 individual worms in both LD and DD showed significantly higher similarity for matching initial/repetition runs. There were also a few cases where an initial run was significantly less similar to the matching repetition run than to the control group (LD: 3/13, DD: 2/13). NAP strain individuals (**A**) showed little behavioral reproducibility with some worms showing significantly higher similarity for

matching initial/repetition runs (LD: 1/5, DD: 2/4), while other worms it was significantly lower (LD: 2/5, DD: 2/4).
(JPG)

**S5 Fig. Worm behavior reproducibility quantified via binary activity overlap approach.** Related to Figs 1E and S4. Similarity (overlap activity scores) of initial runs against matching repetition runs, and against mean values of all other (nonmatching) repetition runs were compared via Wilcoxon matched-pairs signed-rank test for LD (**A**) and DD (**B**, gray background). Significance levels: $^*p < 0.05$, $^{**}p < 0.01$, $^{***}p < 0.001$.
(JPG)

**S6 Fig. Individual actograms of worms characterized for RNASeq analysis.** Related to Fig 2. Double-plotted actograms of individual PIN wild-type worms are shown. Locomotor activity was recorded over 3 d of LD (16 h:8 h) and 3 d of DD. Per behavioral recording, 25 worms were investigated in parallel. Two identical behavior chambers were used for recordings. Each page contains all worms of a characterization run including worms that matured and were thus excluded. #: individual worm identifier. Letters indicate characterization as rhythmic (R), arrhythmic (A), or intermediate (i).
(PDF)

**S7 Fig. Magnified individual actograms of arrhythmic worms for RNASeq analysis.** Related to Figs 2 and S6. Double-plotted actograms of individual PIN wild-type worms are shown. Locomotor activity was recorded over 3 d of LD (16 h:8 h) and 3 d of DD. Y-axis is magnified 10-fold relative to S6 Fig to better illustrate activity patterns. (**A**) Worms characterized as arrhythmic. (**B**) Worms that crawled out and where no activity could be recorded. Red shading indicates that worms crawled out of the tracking well. For rec #4, worm #16, the animal left the tracking well before the start of recording but occasionally moved its head into the well, which was then tracked (panel **C**). This happened mostly when the worm was most active, i.e., during the dark phases. This comparison illustrates that although arrhythmic worms showed overall lower activity, they were far from inactive. Recording numbers and individual worm identifiers (#) match those in S6 Fig.
(PDF)

**S8 Fig. Differential transcripts cycling in behaviorally rhythmic vs.** arrhythmic worms. Related to Fig 3. Representative transcripts with significant 24-h cycling only rhythmic worms (**A**) or only in arrhythmic worms (**B**) were selected at random from the bottom of the heatmaps in Fig 3B and 3D and are plotted here (rhythmic: blue, arrhythmic: red). Bold *p*-values (FDR-corrected) indicate significant 24-h cycling. Per time point, $n = 3$ replicates were measured. (**C**) Transcripts with significant 24-h cycling only in behaviorally rhythmic worms. The used transcripts ($n = 26$) are associated with the GO-terms "neuromuscular process controlling balance," "axon regeneration," "visual behavior," and "response to hypoxia." (**D**) Transcripts with 24-h cycling only in behaviorally arrhythmic worms. The used transcripts ($n = 25$) are associated with the GO-terms "fatty acid beta-oxidation using acyl-CoA dehydrogenase," "glucose metabolic process," "excretion," "response to vitamin A," "mitochondrial transmembrane transport," and "phosphatidylinositol phosphorylation." Mean transcript SDs for a given transcript and phenotype were calculated as mean of the SDs for the 6 individual time points ($n = 3$ samples per time point). Variance was compared between rhythmic (blue) and arrhythmic (red) phenotypes via paired 2-sided *t* test. Black lines indicate value pairs belonging to the same transcript. Significance levels: $^*p < 0.05$, $^{**}p < 0.01$, $^{***}p < 0.001$, $^{****}p < 0.0001$. While there was a general trend of lower variance in the phenotype with cycling, in both (**C**) and (**D**), there are transcripts with no or the opposite trend. The higher variances for

neuronal/behavioral transcripts (**C**) in behaviorally arrhythmic worms are fully consistent with the observed behavior. Detailed RAIN results are provided in S2E–S2G Table.
(JPG)

**S9 Fig. Immunohistochemistry of *Pdu*-PDF in *Platynereis'* wild-type and pdf mutant heads.** Related to Fig 5. Heads of immature worms were stained with anti-*Pdu*-PDF antibody, in combination with Hoechst staining of nuclei. (**A**) Schematic of *P. dumerilii* head. (**B**, **B'**) *pdf* +/+ (wild-type) heads show staining between the posterior eyes partially overlapping with the posterior oval-shaped domain (arrows) between the posterior eyes (p.e.) [37], adjacent to the anterior eyes (a.e. arrowheads), as well as in neural projections in the center of the head and mushroom bodies (*). (**C-E'**) Immunohistochemistry with anti-*Pdu*-PDF in worms carrying different combinations of mutant *pdf* alleles (Δ−14/Δ−14; Δ+4/Δ+4; Δ−14/Δ+4) showed that neither of the mutant worms have PDF staining. Scale bar: 100 μm.
(JPG)

**S10 Fig. Individual worm actograms of *pdf* wild type/mutant comparison.** Shown are double-plotted actograms of mixed VIO/PIN background *pdf* wild types (**A**) and mutants (**B**) related to Fig 6, the initial VIO background wild types (**C**) and mutants (**D**) related to S11 Fig, and VIO backcrossed wild types (**E**) and mutants (**F**) related to S12 Fig. Locomotor activity was recorded over 4 d of LD (16 h:8 h) and 4, 5, or 8 d of DD. #: individual worm identifier. Genotypes of *pdf* mutants (−14/−14, +4/+4, −14/+4) are indicated at the bottom of the respective actograms. Red shading indicates that worms crawled out of the tracking well. Worms that were excluded from statistics due to maturation during or within 1 week after the recording are not shown, as maturation strongly alters their overall behavior.
(PDF)

**S11 Fig. Behavioral rhythmicity of *pdf* wild types and mutants in the initial VIO strain background.** Related to Fig 6. In 4 recordings, $n = 34$ *pdf* wild types from 4 mating batches were compared to $n = 43$ *pdf* mutants from 5 mating batches (−14/−14 $n = 33$, −14/+4 $n = 10$). (**A**) Circadian locomotor activity of VIO strain *pdf* wild types (black) and mutants (red) under 4 d of LD and 8 d of DD. Individual worm actograms are provided in S10C and S10D Fig. (**B**) Cumulative activity over the early day (0–8), late day (8–16), and night (16–24) in LD and DD. (**C**, **D**) Period/power of wild type and mutant locomotor rhythms in the circadian range (20 h–28 h) in LD and DD determined by Lomb–Scargle periodogram. Statistical differences (panels **B**–**D**) were determined via Mann–Whitney U-test. Significance levels: $^*p < 0.05$, $^{**}p < 0.01$, $^{***}p < 0.001$, $^{****}p < 0.0001$. For period/power values, see S1D Table. For further info on figure labeling, see Fig 1. The reduced rhythmicity of *pdf* mutants disappeared directly after an outcross against the PIN strain (Fig 6) and could not be recovered by backcross against the VIO strain (S12 Fig). Hence, we consider it not causally connected to the *pdf* mutant locus. We attribute the difference of the VIO wild types here compared to the VIO strain worms in Fig 1 to the high polymorphism rate, which in marine invertebrates can reach 4%–5% in the maternal vs. paternal genomes of ONE individual [110].
(JPG)

**S12 Fig. Behavioral rhythmicity of *pdf* wild types and mutants after backcross into the VIO strain.** Related to Fig 6. In 1 recording, $n = 19$ *pdf* wild types from 5 mating batches were compared to $n = 19$ *pdf* mutants from 6 mating batches (−14/−14 $n = 9$, +4/+4 $n = 10$). (**A**) Circadian locomotor activity of VIO-backcrossed *pdf* wild types (black) and mutants (red) under 4 d of LD and 8 d of DD. Individual worm actograms were are provided in S10E and S10F Fig. (**B**) Cumulative activity over the early day (0–8), late day (8–16), and night (16–24) in LD and DD. (**C**, **D**) Period/power of wild type and mutant locomotor rhythms in the circadian range

(20 h–28 h) in LD and DD determined by Lomb–Scargle periodogram. Statistical differences were determined via Mann–Whitney U-test (panels **B** and **C**) or unpaired 2-sided *t* test (panel **D**). Significance levels: *$p < 0.05$, **$p < 0.01$, ***$p < 0.001$, ****$p < 0.0001$. For period/power values, see S1E Table. For further info on figure labeling, see Fig 1. The persistence of the phenotype observed in VIO/PIN background worms (Fig 6) reinforces that this is a solid *pdf* mutant phenotype and that the initial rhythmicity reduction in the VIO strain (S11 Fig) is not causally connected to the *pdf* mutant locus.
(JPG)

**S1 Table. Rhythm analysis of worm used in behavior experiments for strain comparison, RNAseq characterization, and *pdf* wild type/mutant comparisons.** Related to Figs 1, 2, 6, S2, S6, S10 and S11.
(XLSX)

**S2 Table. RNAseq expression counts and RAIN rhythm analysis of behaviorally rhythmic and arrhythmic PIN wild-type worms.** Related to Figs 2, 3 and S8.
(XLSX)

**S3 Table. GO-term enrichment of transcripts cycling in rhythmic and arrhythmic worms.** Related to Fig 3.
(XLSX)

**S4 Table. Peptides used for prepro-PDF and PDFR phylogenetic tree construction.** Related to Fig 4.
(XLSX)

**S5 Table. PDFR deorphanization measurement data.** Related to Fig 4.
(XLSX)

**S6 Table. *pdf* wild-type and mutant worm core clock gene qPCR data.** Related to Fig 5.
(XLSX)

**S7 Table. LD/DD comparison of cumulative locomotor activity in *pdf* wild-type and mutant worms.** Related to Fig 6.
(XLSX)

## Acknowledgments

We are further grateful to Andrij Belokurov, Margaryta Bosysova, and Netsaneh Getachew (Aquatic Facility) for worm cultures and routine genotyping support.

## Author Contributions

**Conceptualization:** N. Sören Häfker, Liliane Schoofs, Florian Raible, Kristin Tessmar-Raible.

**Data curation:** N. Sören Häfker, Laurenz Holcik, Alexander W. Stockinger.

**Formal analysis:** N. Sören Häfker, Laurenz Holcik, Audrey M. Mat, Aida Ćorić, Karim Vadiwala, Isabel Beets, Alexander W. Stockinger, Carolina E. Atria, Stefan Hammer, Roger Revilla-i-Domingo, Kristin Tessmar-Raible.

**Funding acquisition:** N. Sören Häfker, Florian Raible, Kristin Tessmar-Raible.

**Investigation:** N. Sören Häfker, Laurenz Holcik, Audrey M. Mat, Aida Ćorić, Karim Vadiwala, Isabel Beets, Stefan Hammer, Florian Raible.

**Project administration:** Florian Raible, Kristin Tessmar-Raible.

**Resources:** Liliane Schoofs, Kristin Tessmar-Raible.

**Software:** Laurenz Holcik.

**Supervision:** Roger Revilla-i-Domingo, Liliane Schoofs, Florian Raible, Kristin Tessmar-Raible.

**Validation:** N. Sören Häfker, Audrey M. Mat.

**Visualization:** N. Sören Häfker, Aida Ćorić.

**Writing – original draft:** N. Sören Häfker, Kristin Tessmar-Raible.

**Writing – review & editing:** Laurenz Holcik, Audrey M. Mat, Aida Ćorić, Karim Vadiwala, Isabel Beets, Alexander W. Stockinger, Carolina E. Atria, Stefan Hammer, Roger Revilla-i-Domingo, Liliane Schoofs, Florian Raible.

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
