## [Editor Report · Decision Letter 0]

8 Jan 2024

Dear Kristin, 

Thank you for submitting your revised manuscript entitled "Behaviorally rhythmic and arrhythmic individuals of a marine annelid exhibit similar levels of molecular rhythmicity in the brain" for consideration as a Research Article by PLOS Biology, and please accept my apologies for the delay in getting back to you. The PLOS Biology office was closed over the holidays, which caused the delay. 

I am writing to let you know that I have now had a chance to discuss your revision with the Academic Editor, and we would like to send your submission back to the original reviewers for their input. 

Once your full submission is complete, your paper will undergo a series of checks in preparation for peer review. After your manuscript has passed the checks it will be sent out for review. To provide the metadata for your submission, please Login to Editorial Manager (https://www.editorialmanager.com/pbiology) within two working days, i.e. by Jan 10 2024 11:59PM.

Kind regards,

Luke

Lucas Smith, Ph.D.

Senior Editor

PLOS Biology

lsmith@plos.org

---

## [Decision Letter · Decision Letter 1]

9 Feb 2024

Dear Kristin,

Thank you for your patience while we considered your revised manuscript "Behaviorally rhythmic and arrhythmic individuals of a marine annelid exhibit similar levels of molecular rhythmicity in the brain" for publication as a Research Article at PLOS Biology. This revised version of your manuscript has been evaluated by the PLOS Biology editors, the Academic Editor and by the original reviewers 2 and 3.

As you will see in their comments below, both reviewers agree that the study has been strengthened in the revision. Reviewer 2 has suggested that we accept the study, after relatively minor revisions, while Reviewer 3 has lingering concerns about whether the behavioral data is robust enough to be used for rhythmic analyses. While we do appreciate this reviewer's concerns, after discussing this point with the Academic Editor, and considering the additional work that was done in this revision, our opinion is that, on balance, the data provided is strong enough for publication and we would not require additional data or analyses to address these last concerns. 

Therefore, based on the reviews and on our Academic Editor's assessment of your revision we are likely to accept this manuscript for publication, provided you satisfactorily address the remaining points raised by reviewer 2, and possibly by adding more discussion in response to reviewer 3. As a last note about the reviews, reviewer 3 has also suggested that it may be incorrect to state that arrhythmic behavior is adaptive because it is beneficial for the species, noting that selection is driven by individual fitness, not the species. We would encourage you to consider and carefully respond to this concern. After discussing this point with my colleagues, I would add that there may be instances where evolution can select for between-individual differences in a population (e.g. bet-hedging, balancing selection, etc.) - and so it may be interesting to discuss this evolutionary context a bit further, if you feel that is relevant. 

**IMPORTANT: As you address the last reviewer comments, please also address the following editorial requests

1) TITLE: We appreciate that the reviewers asked for the previous title to be toned down, but we think the current version could be refined a bit further to be more enticing to our broad readership. If you agree, (and if supported) we suggest it be changed to something like:

"Molecular circadian rhythms are robust in marine annelids lacking rhythmic behavior"

2) DATA: Thank you for providing the data underlying your figures as supplemental tables and as a deposition to dryad. For the most part these datasets meet our criteria for publication however I have a few minor requests: 

a. I see that Table 2 is provided as a dropbox link. However to ensure the permanence of this dataset, we ask that you upload it along with your manuscript. If it proves to big to be uploaded, we would ask that you put it on a data repository. (Please feel free to loop me in if you have questions about this request) 

b. For the data in the supplemental tables, can you more clearly spell out which how the data in each tab relates to specific figure panels. I saw those details present in some of your supplemental tables, but in others it was a bit less clear. 

c. Please add a sentence to each figure legend detailing where the underlying data can be found. For example, you can add the sentence "the data underlying this figure can be found in supplemental table __" 

3) CODE: Per journal policy, if any code was generated to support the conclusions of your manuscript, we would require that you make it available without restrictions upon publication. Please ensure that any code is sufficiently well documented and reusable, and that your Data Statement in the Editorial Manager submission system accurately describes where your code can be found. (I see that you do detail the codes that you used in your resources section, which is great. I am just adding this note as a precaution in case you had any newly generated code not included there).

We expect to receive your revised manuscript within two weeks. 

*Published Peer Review History*

*Press*

Sincerely,

Luke

Lucas Smith, Ph.D.

Senior Editor

lsmith@plos.org

PLOS Biology

Reviewer remarks:

Reviewer #2: The authors have extensively revised and improved their very interesting manuscript. My major concerns have been addressed convincingly and thoroughly. I have a few suggestions for the authors to consider:

1) There is no doubt that there are important phenotypic variations between strains on figure 1. However, as I mentioned in my first review, the same strain can behave rather differently over time (S11). Therefore, I wonder whether the variations seen between strains on figure 1 are really strain-specific, or if they reflect, at least in part, the state of the colonies at a specific time. It seems difficult to know for sure at this point, so it would be worth bringing this up in the result or discussion section.

2) Really striking how, on figure S5, the high reproducible behavior are lost in the control pairing. I had noted in the previous review that it seemed the highly reproducible behaviors were frequently "hypoactive" animals. The authors have now convincingly established these animals are active throughout, but do not show a nocturnal peak of activity, and importantly do not show any sign of sickness. It would be interesting to determine whether indeed this lack of nocturnal behavior is enriched in the most reproducible phenotypes, or perhaps even drive most of the reproducibility. It might be the most stable behavior state. It might also be worth mentioning the supplemental figures magnifying the activity of the arrhythmic, low activity animals earlier in the text. 

3) Line 201: Rephrase : "To closer investigate" to " To investigate more closely ".

4) Line 84-5. The statement here is worded too strongly in my opinion, though the data in rev 15 certainly suggest such a model. 

Reviewer #3: 

The revised manuscript addresses many of the comments that I made on the original version. However, a couple of issues have not been resolved. My major point #3 was that the different patterns of behavior (e.g., uni- vs bimodal profile) may affect the spectral analysis and, therefore, are not comparable. Furthermore, some of the actograms in Figure 1E are just blank graphs with no activity. This low level of activity is not suitable for rhythm analysis.

The other point that I have raised is about the suggestion that arrhythmic behavior is adaptive because it is beneficial for the species (this suggestion is also present in the abstract). However, selection is driven by individual fitness, not the species.

---

## [Editor Report · Decision Letter 2]

29 Feb 2024

Dear Kristin,

Thank you for the submission of your revised Research Article "Molecular circadian rhythms are robust in marine annelids lacking rhythmic behavior" for publication in PLOS Biology and thank you for addressing the last reviewer comments and editorial requests in this revision. On behalf of my colleagues and the Academic Editor, Martha Merrow, I am pleased to say that we can in principle accept your manuscript for publication, provided you address any remaining formatting and reporting issues. These will be detailed in an email you should receive within 2-3 business days from our colleagues in the journal operations team; no action is required from you until then. Please note that we will not be able to formally accept your manuscript and schedule it for publication until you have completed any requested changes.

PRESS

Sincerely, 

Luke

Lucas Smith, Ph.D.

Senior Editor

PLOS Biology

lsmith@plos.org